# When Vision Needs a Second Look: Tool-Augmented Active Perception for Earth Observation

## Abstract

Earth Observation (EO) uses satellite and aerial imagery to monitor the Earth's surface, supporting critical applications in infrastructure, agriculture, and climate change. As governments and industry scale EO pipelines, reliable automation has become essential. Yet, current Vision-Language Models are limited to coarse-grained perception, struggling to execute the precise, multi-step reasoning required for operational decision-making. Recent evaluations on benchmarks like GeoBench-VLM highlight this shortcoming: even state-of-the-art models show low accuracy and frequently struggle with tasks requiring precise numerical reasoning and domain-specific knowledge, such as object counting, crop-type classification, and assessing vegetation health. These limitations stem from their static, monolithic inference pipeline, which prevents adaptive analysis and error correction. To address these limitations, we introduce *GeoScout-Agent*, an autonomous agentic framework designed to overcome these constraints by coupling GPT-5-mini's tool capabilities. The system built upon LangChain dynamically invokes code execution, progressive zooming, sharpening, and external context verification, while DINOv3 and SAM3 provide zero-shot segmentation and high-quality feature extraction for richer LLM context. This coordinated framework enables the model to iteratively decompose, validate, and refine its predictions rather than relying on a single forward pass. Evaluated on GeoBench-VLM, our approach achieves substantial gains over standard VLM baselines. *GeoScout-Agent* consistently resolves intermediate failures, improves geospatial understanding, and achieves a relative 17.3% improvement across the evaluated tasks over the baseline approach. We will publicly release our code upon acceptance.

## 1 Introduction

Geospatial reasoning is inherently an active and iterative process. Human analysts do not interpret satellite imagery through a single static view; instead, they dynamically zoom into regions of interest, enhance visual contrast, isolate object clusters, and cross-check local evidence before arriving at a decision. In contrast, most contemporary vision-language models (VLMs) operate in a passive, single-pass manner. They consume a fixed-resolution image and produce an answer without the ability to manipulate visual input or verify intermediate hypotheses. This limitation becomes particularly severe in Earth observation, where small objects such as vehicles and rooftops, environmental artifacts including haze and shadows, and subtle spatial cues frequently determine the correct interpretation.

Recent benchmarks such as GeoBench-VLM Danish et al. (2025) make this limitation of single-pass VLM inference in geospatial reasoning explicit. Even state-of-the-art systems, including LLaVA-OneVision (Li et al., 2024), plateau at roughly 41% accuracy and exhibit systematic failures in tasks that require numerical precision, multiscale inspection, or spatial localization such as Building Counting, Vehicle Counting, Crop Classification, and Tree Health Assessment. These errors are not merely a consequence of model capacity, but stem from the rigidity of monolithic inference pipelines that prevent adaptive analysis and error correction.

At the same time, the computer vision community has produced powerful domain-specific perception models, including foundation encoders and segmentation systems tailored to remote sensing imagery. Models such as

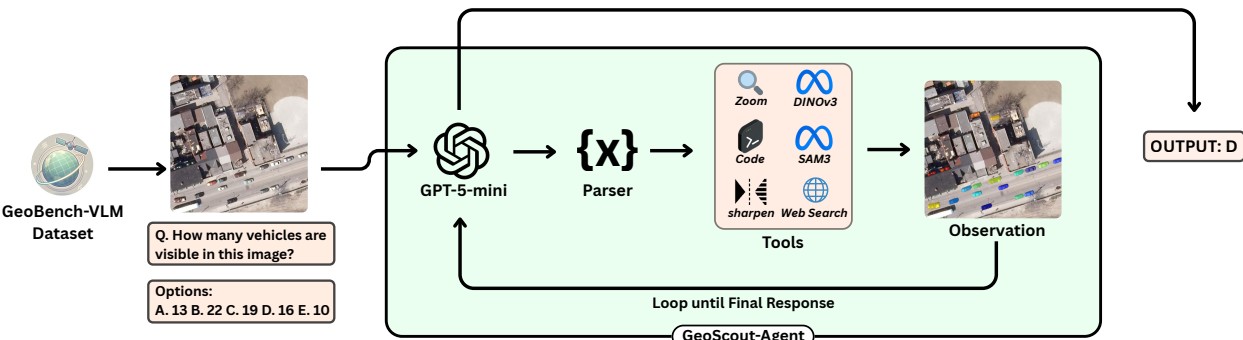

Figure 1: Framework overview of GeoScout-Agent, a tool-augmented Earth Observation (EO) agent. Given a question and a satellite image, GeoScout-Agent iteratively reasons over visual evidence and invokes specialized tools including multiscale zoom, image sharpening, DINOv3-based unsupervised object discovery, SAM3 zero-shot segmentation, code execution, and web-based context verification. Tool outputs are fed back into the reasoning loop, enabling progressive hypothesis refinement and error correction until a final, validated prediction is produced.

DINOv3 (Siméoni et al., 2025) and SAM3 (Carion et al., 2025) offer strong zero-shot detection and instance-level reasoning capabilities, yet they are typically deployed as standalone predictors or offline preprocessing steps. As a result, their potential to support iterative, hypothesis-driven reasoning within VLMs remains largely untapped. Bridging this gap requires rethinking geospatial reasoning as a closed-loop process in which perception modules act as controllable subroutines rather than static feature extractors.

Rather than introducing a new vision backbone, this work studies Earth observation reasoning in a closed-loop perception-action setting over raw imagery. Within this setting, the model can iteratively inspect image regions, invoke specialized tools, and refine its prediction based on intermediate evidence. This formulation is particularly relevant for EO tasks involving small objects, dense scenes, and subtle spatial cues that are often challenging for single-pass inference.

We instantiate this idea in *GeoScout-Agent*, an agentic Earth Observation framework that operationalizes active perception for geospatial reasoning. Our system integrates a base vision–language model, GPT-5-mini (OpenAI, 2025), with a graph-based agent controller built on LangChain (Topsakal & Akinci, 2023), enabling the model to iteratively inspect imagery, invoke specialized tools, and refine its predictions based on intermediate visual evidence. The agent is equipped with a suite of EO-relevant tools, including multiscale zoom for adaptive resolution control, image sharpening for degraded satellite data, DINOv3-based unsupervised visual discovery, SAM3-based zero-shot segmentation for region-level reasoning, Python code execution for deterministic image analysis, and web-based retrieval for contextual verification. By embedding these tools within a unified perception-reasoning loop, the agent moves beyond passive observation toward deliberate, verifiable geospatial inference. Our main contributions are as follows:

- We propose *GeoScout-Agent*, an agentic framework for Earth observation that transforms geospatial reasoning from single-pass VLM inference into an iterative perception-action loop with tool-augmented analysis.

- We integrate multiple specialized vision and analysis tools as controllable subroutines within a language-driven agent, enabling multiscale inspection, segmentation-assisted counting, and iterative refinement.

- We demonstrate consistent and substantial performance gains on GeoBench-VLM, particularly for dense object counting and spatial reasoning tasks, highlighting the limitations of passive VLMs and the benefits of active perception.

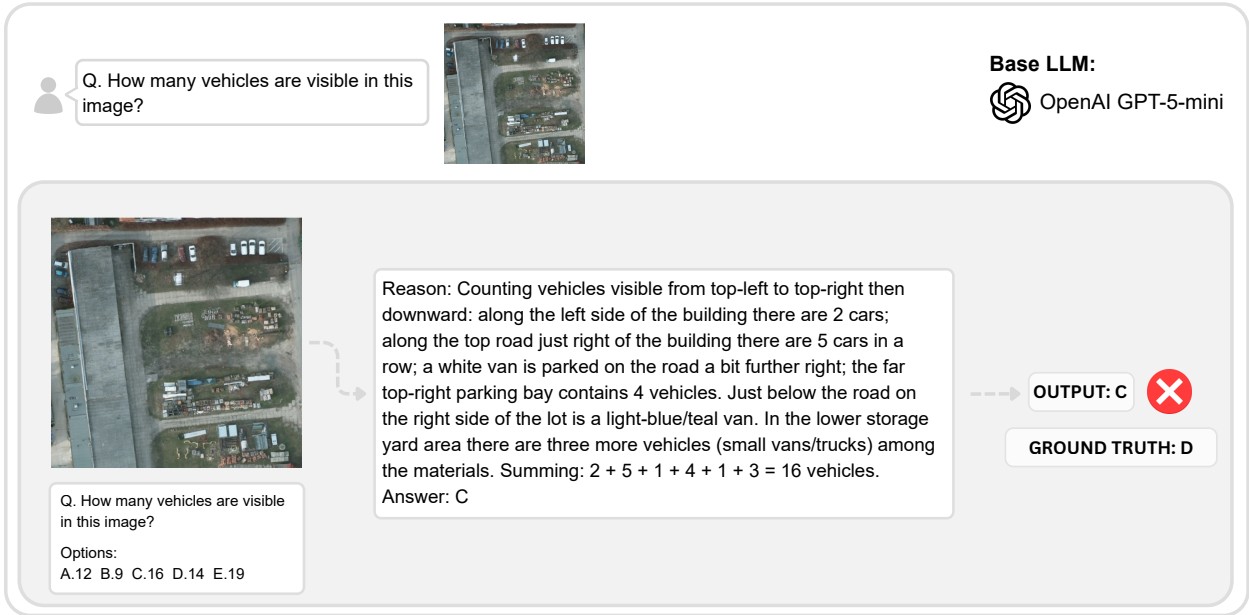

Figure 2: Failure case of the base GPT-5-mini model on a dense vehicle counting task. Despite generating a plausible and internally consistent reasoning trace, the model produces an incorrect count due to occlusion, object adjacency, and limited visual resolution, highlighting the brittleness of single-pass inference without explicit visual verification.

## 2 Related Work

### 2.1 Vision-Language Models for Earth Observation

Recent work has adapted general-purpose vision-language models to Earth observation through fine-tuning, dataset curation, or multi-sensor alignment. Systems such as GeoChat (Kuckreja et al., 2024), LHRS-Bot (Muhtar et al., 2024), EarthDial (Soni et al., 2025), and SkySense (Luo et al., 2024) enable natural language interaction with satellite imagery and improve accessibility for downstream users. While effective for coarse scene understanding and captioning, these approaches largely operate under a single forward-pass inference paradigm. As a result, they struggle with tasks that require numerical precision, dense object counting, or spatial verification, limitations that are systematically exposed by GeoBench-VLM evaluations.

Parallel to VLM development, the computer vision community has produced strong foundation models for remote sensing perception. Models such as DINOv3 (Siméoni et al., 2025) provide robust unsupervised visual representations, while segmentation frameworks like SAM3 (Carion et al., 2025) enable high-quality zero-shot instance masks across domains. While these specific models are recent, prior EO pipelines have employed similar representation learning and segmentation models primarily as static preprocessing components or independent predictors (Xiao et al., 2025). Their outputs are consumed once, without allowing a language model to adaptively invoke, refine, or cross-check visual evidence. In contrast, our work integrates these perception models as callable tools within an agentic loop, allowing the language model to selectively apply segmentation, discovery, and analysis in response to intermediate failures.

### 2.2 Agentic Vision-Language Approaches for Earth Observation

More recent efforts explore agent-based formulations for EO reasoning, aiming to decompose complex geospatial tasks into sequences of tool calls. GeoFlow (Bhattaram et al., 2025) proposes workflow automation through predefined execution graphs optimized for efficiency, but lacks adaptive visual feedback during inference. Remote Sensing ChatGPT (Guo et al., 2024) and related systems employ LLM-driven planners to

orchestrate external remote sensing models, with visual information primarily injected via captions and tool outputs rather than end-to-end image-driven reasoning.

Several domain-specific agents target narrowly scoped EO tasks. TreeGPT (Du et al., 2023) focuses on forestry applications such as tree segmentation and ecological parameter estimation, while Change-Agent (Liu et al., 2024a) addresses bi-temporal change detection through instruction-following workflows. GeoMap-Agent (Huang et al., 2025) targets geological mapping with structured visual inputs. Although effective within their respective domains, these systems rely on task-specific templates or qualitative reasoning and do not support general-purpose, iterative visual verification across heterogeneous EO tasks.

RS-Agent (Xu et al., 2024) represents the closest conceptual precursor to our work. It introduces an LLM-centered controller that routes queries to external tools and retrieval systems. However, its primary emphasis lies in knowledge retrieval and decision routing, rather than visual grounding. While RS-Agent possesses image processing tools, visual inputs are not dynamically manipulated for iterative verification, and intermediate perception errors cannot be corrected once a tool call is made. In contrast, *GeoScout-Agent* treats vision as an explicit component within an iterative reasoning loop, allowing multiscale zooming, sharpening, segmentation, and code-based analysis to be invoked as needed during inference. Building on prior agentic vision-language work, we present a fully instantiated EO system that performs closed-loop perception, verification, and reasoning directly over raw satellite imagery.

## 3 Methodology

### 3.1 Framework Architecture

We formalize the geospatial reasoning process as a graph-based agent framework within the LangChain ecosystem (Chase, 2022). The architecture consists of a directed graph containing two primary nodes:

**Reasoning Node (Agent):** A GPT-5-mini instance that analyzes the current AgentState, reasoning over the conversation history and visual evidence to generate tool calls or a final response.

**Execution Node (Tools):** A runtime environment that parses tool calls, executes the corresponding Python routines or model inference (e.g., SAM3, DINOv3), and appends the outputs to the state.

The AgentState is implemented as an ordered message list that persistently accumulates all user inputs, agent actions, and tool outputs. This additive state representation preserves the full interaction trace across iterations, enabling *GeoScout-Agent* to reason over prior visual evidence, failed attempts, and intermediate computations rather than relying only on the most recent observation.

### 3.2 Vision Language Controller

The central decision engine of the system is a GPT-5-mini model configured as the orchestrator (Fig 1). At each iteration, *GeoScout-Agent* receives the full AgentState and decides whether additional evidence is required. It then either invokes one or more tools or produces a final answer. Tool outputs are appended to the state, allowing subsequent reasoning steps to condition explicitly on intermediate perceptual and computational results.

### 3.3 Visual Perception and Dynamic Analysis Tools

To support fine-grained geospatial reasoning, *GeoScout-Agent* is equipped with a modular set of vision tools that combine high-capacity neural models with lightweight image manipulation and numerical analysis utilities. These tools operate independently of the VLM and can be invoked at any time. Crucially, visual artifacts produced by any tool (e.g., masks, crops, enhanced views) are recursively appended to the multimodal context, enabling direct reasoning over the transformed state.

**Specialized Vision Modules** We integrate two vision foundation models as specialized perceptual components.

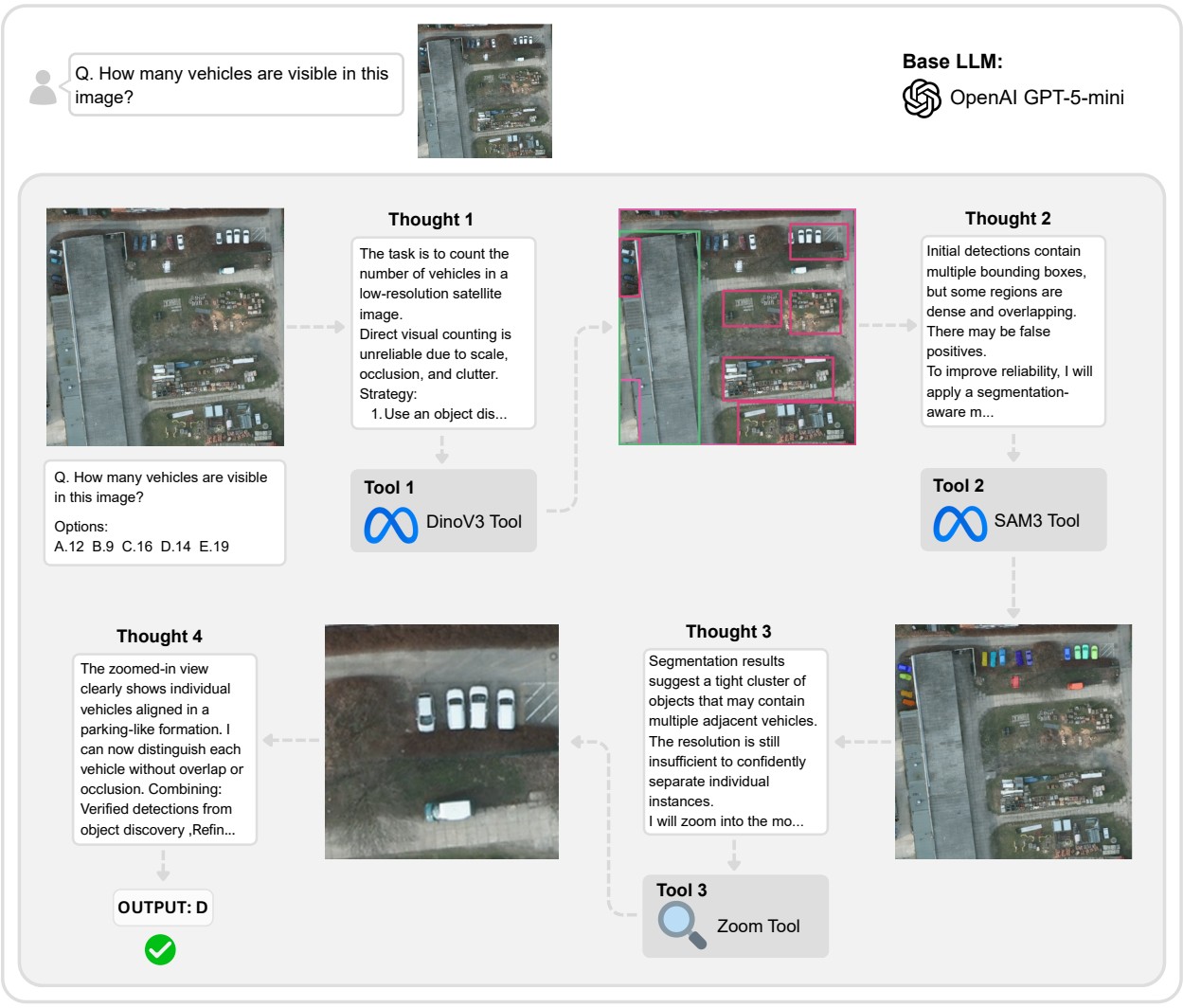

Figure 3: Successful vehicle counting with GeoScout-Agent. The agent incrementally refines its hypothesis through object discovery, segmentation, and targeted zooming, enabling reliable instance separation in a cluttered scene and yielding the correct final prediction via iterative visual verification.

**SAM3 Instance Segmentation.** Given an image and a textual query, SAM3 produces zero-shot instance masks with associated confidence scores, enabling object counting, region isolation, and pixel-level analysis.

**DINOv3 Unsupervised Detection.** DINOv3 performs class-agnostic object discovery by clustering dense visual features and extracting bounding boxes for salient regions, supporting exploratory analysis when object categories are unknown.

**Image Manipulation and Analysis Tools** *GeoScout-Agent* can also invoke procedural tools for targeted visual refinement:

**Python Code Execution.** A flexible execution interface that allows deterministic image analysis and transformation, including pixel statistics, geometric operations, and custom filters using OpenCV-based Python code execution.

**Zoom and Sharpen.** The zoom tool performs adaptive field-of-view control by cropping image regions around agent-selected normalized coordinates and resizing them to a fixed resolution, enabling focused

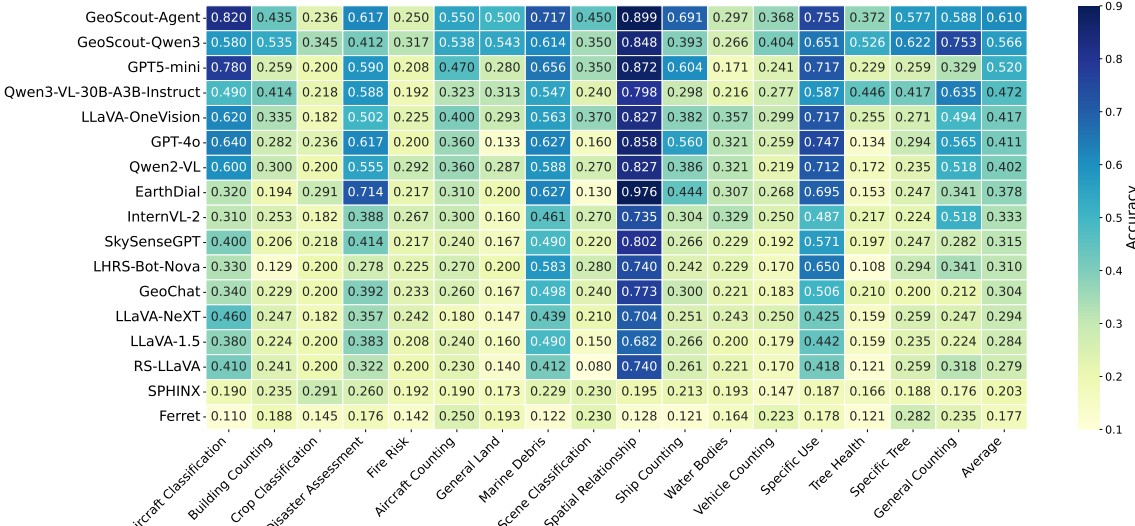

Figure 4: Task-level accuracy heatmap of vision–language models on GeoBench-VLM. Tool-augmented inference enables substantial gains over single-pass baselines, particularly for object counting, scene understanding, and spatial reasoning, demonstrating the impact of closed-loop visual analysis.

inspection of small or dense structures. The sharpen tool applies an unsharp mask to enhance edges and contrast in degraded or hazy imagery.

**Web Search.** *GeoScout-Agent* may invoke web retrieval only in cases where visual evidence alone is insufficient and the task requires external factual knowledge. This is most relevant for fine-grained categories whose correct identification depends partly on world knowledge, such as distinguishing ship classes with similar visual structure but different class-specific attributes. Web search is therefore an auxiliary and selectively used tool rather than a core component of the framework; the primary contribution of GeoScout-Agent remains iterative visual reasoning over imagery.

## 3.4 Inference Workflow

Inference proceeds through an iterative perception-reasoning loop defined by the LangGraph cycle. At each iteration, *GeoScout-Agent* evaluates the current state, determines whether additional evidence is required, and either invokes tools or generates a final response. Tool outputs are appended to the state and incorporated into subsequent reasoning steps.

Termination is governed by explicit control policies. The loop terminates if any of the following conditions is satisfied: (1) The agent produces a final answer without requesting further tools, (2) A predefined maximum number of tool calls is reached (fixed to 3 across all experiments), or (3) An empty response is detected.

In cases (2) and (3), forced synthesis is triggered, wherein the LLM is invoked without tool bindings to generate a final textual answer. The graph is additionally compiled with a recursion limit of 20 to prevent unbounded execution. These safeguards ensure bounded computation, prevent infinite loops, and guarantee that inference always yields a textual prediction.

## 4 Experiments and Results

### 4.1 Experiments Across Task Categories

Beyond the standard GeoBench-VLM suite (Danish et al., 2025), we evaluate four additional models to isolate the impact of our agentic framework. We employ GPT-5-mini and Qwen-3-VL-30B-A2B-Instruct (Bai et al.,

2025) as our primary tool-free baselines. We then compare these against their respective enhanced counterparts: GeoScout-Agent (GPT-5-mini) and GeoScout-Qwen3 (Qwen-3-VL). For both GeoScout variants, the base models are augmented with our agentic framework, providing access to a suite of tools to facilitate iterative visual inspection and verification. Example tasks from all categories are shown in Appendix E.

All experiments were conducted on a single NVIDIA A6000 GPU. For GeoScout-Agent we used GPT-5-mini as the language model with a temperature of 0.3 to ensure consistent and deterministic outputs. For segmentation, SAM3 was configured with a threshold of 0.5. Object detection was handled by DINOv3 with a feature dimension of 518, patch size of 14, and 4 clusters, using a box threshold of 0.02 and a fixed random state of 42 for reproducibility. Image processing tools were set with a sharpening intensity of 2.0, radius of 2, and threshold of 3, along with a zoom factor of 2.0 and an output size of 448. A complete summary of all hyperparameters is provided in Table 8 in Appendix B.

**Scene Understanding:** Scene understanding tasks span agriculture, urban planning, and environmental monitoring. These tasks are challenging due to subtle inter-class differences and limited spatial resolution. Compared to the tool-free baseline, *GeoScout-Agent* consistently improves performance, outperforming all single-pass VLMs on this category (Table 1). This gain indicates that iterative inspection and targeted visual refinement are particularly beneficial for resolving fine-grained scene-level distinctions.

**Object Classification:** Object classification tasks, such as identifying ships and aircraft, are central to maritime surveillance and airspace monitoring. *GeoScout-Agent* achieves a 10.84% relative improvement over the baseline, demonstrating that tool-augmented reasoning improves object-type separation in visually similar categories, where single-pass inference often fails.

Table 1: VLMs accuracies across geospatial tasks. Evaluation includes event detection, object classification, counting, scene understanding, and image captioning.

| Model | Event Det | Object Cls | Counting | Scene Und | Image Cap |
|---|---|---|---|---|---|
| GPT-4o (Hurst et al., 2024) | 0.4726 | 0.5863 | 0.3965 | 0.7114 | 0.6418 |
| EarthDial (Soni et al., 2025) | **0.5418** | 0.4039 | 0.3626 | 0.7705 | 0.5378 |
| Qwen2-VL (Wang et al., 2024b) | 0.4640 | 0.4560 | 0.4019 | 0.6761 | 0.5895 |
| LLaVA-OneVision (Li et al., 2024) | 0.4063 | 0.4593 | 0.4377 | 0.6636 | 0.6317 |
| SkySenseGPT (Luo et al., 2024) | 0.3458 | 0.3094 | 0.3119 | 0.6205 | 0.6416 |
| InternVL-2 (Chen et al., 2024) | 0.3458 | 0.3062 | 0.3280 | 0.5727 | 0.5968 |
| GeoChat (Kuckreja et al., 2024) | 0.3372 | 0.3127 | 0.2922 | 0.6091 | 0.4395 |
| LHRS-Bot-Nova (Li et al., 2025) | 0.2594 | 0.2704 | 0.3286 | 0.6330 | 0.6275 |
| LLaVA-NeXT (Liu et al., 2024b) | 0.3170 | 0.3192 | 0.2737 | 0.5477 | 0.6293 |
| LLaVA-1.5 (Liu et al., 2023) | 0.3228 | 0.3029 | 0.2618 | 0.5625 | 0.6346 |
| RS-LLaVA (Bazi et al., 2024) | 0.2795 | 0.3094 | 0.2534 | 0.5534 | 0.5604 |
| SPHINX (Lin et al., 2023) | 0.2363 | 0.2052 | 0.1860 | 0.2170 | 0.6451 |
| Ferret (You et al., 2023) | 0.1643 | 0.1173 | 0.1956 | 0.1261 | 0.5615 |
| GPT-5-mini (OpenAI, 2025) | 0.4579 | 0.6613 | 0.3971 | 0.7294 | 0.8533 |
| GeoScout-Agent | 0.4901 | **0.7330** | **0.5261** | **0.7728** | **0.8555** |

**Event Detection:** Event detection tasks focus on identifying disasters relevant to environmental monitoring and risk management. Although EarthDial achieves the highest performance in this category, *GeoScout-Agent* consistently outperforms its tool-free counterpart. This improvement indicates that iterative inspection and targeted tool use contribute to more reliable event-level classification.

**Caption Generation:** Image captioning performance is evaluated using BERTScore. *GeoScout-Agent* yields only marginal gains, suggesting that caption quality is primarily determined by the base model's representational capacity rather than iterative visual reasoning. This result is consistent with the fact that captioning emphasizes holistic semantic description rather than fine-grained verification.

**Object Localization and Counting:** This category evaluates the ability to localize and count objects such as buildings, trees, water bodies, vehicles, and aircraft. These tasks are inherently difficult due to heterogeneous scenes, scale variation, and dense object layouts, and are formulated as multiple-choice classification problems.

Prior GeoBench-VLM results show that object counting remains a major failure mode for monolithic VLMs. Both GPT-5-mini and *GeoScout-Agent* perform strongly across counting tasks (Figure 4), with tool augmentation yielding substantial gains. The effect is most pronounced in dense scenes: *GeoScout-Agent* improves building counting by 67.95%, tree counting by 122.78%, and water body counting by 78.72%. Gains are also observed for general aircraft counting (17.02%) and general vehicle counting (78.57%). Fine-grained categories such as specific aircraft and specific vehicle counting improve by 73.68% and 52.70%, respectively, although LLaVA-OneVision remains strongest on specific aircraft counting. Marine debris counting further improves by 28.57%. These results highlight the effectiveness of segmentation-assisted counting for cluttered and overlapping object categories. All reported improvements denote percentage increases over the corresponding tool-free baselines.

Table 2: Referring expression detection results. We report Precision at IoU thresholds 0.5 and 0.25.

| Model | Prec@0.5 | Prec@0.25 |
|---|---|---|
| Sphinx (Lin et al., 2023) | **0.3408** | **0.5289** |
| EarthDial (Soni et al., 2025) | 0.2429 | 0.4139 |
| GeoChat (Kuckreja et al., 2024) | 0.1151 | 0.2100 |
| Ferret (You et al., 2023) | 0.0943 | 0.2003 |
| Qwen2-VL (Wang et al., 2024b) | 0.1518 | 0.2524 |
| GPT-4o (Hurst et al., 2024) | 0.0087 | 0.0386 |
| LHRS-Nova (Li et al., 2025) | 0.0930 | 0.2423 |
| SkySenseGPT (Luo et al., 2024) | 0.1082 | 0.3224 |
| GPT-5-mini (OpenAI, 2025) | 0.0491 | 0.1631 |
| GeoScout-Agent | 0.0756 | 0.2204 |

**Referring Expression Detection:** This is a visual grounding task where the model must localize objects in aerial/satellite imagery based on natural language referring expressions. Unlike multiple-choice QA, the model outputs bounding box coordinates rather than selecting an answer. GPT-5-mini shows lower performance on this task, whereas Sphinx achieves the highest precision at both IoU thresholds (Table 2). Nevertheless, *GeoScout-Agent* consistently improves GPT-5-mini's precision, indicating that explicit segmentation and region-level reasoning partially mitigate localization challenges, even in tasks where overall performance remains limited.

**Temporal Understanding:** Temporal reasoning tasks evaluate changes across time, which are central to remote sensing analysis. We evaluate five temporal tasks: crop classification, damaged building counting, disaster type classification, farm pond change detection, and land use classification ( Table 3). *GeoScout-Agent* consistently improves over GPT-5-mini across all tasks, highlighting the benefit of tool-assisted reasoning for modeling temporal changes.

Table 3: Performance of vision-language models on temporal geospatial tasks. Evaluation covers crop classification, damaged building counting, disaster type classification, farm pond change detection (CD), and land use classification. Results show that GeoScout-Agent consistently improves GPT-5-mini across all temporal tasks.

| Model | Crop Cls | Damaged Bldg Cnt | Disaster Cls | Farm Pond CD | Land Use Cls |
|---|---|---|---|---|---|
| EarthDial (Soni et al., 2025) | **0.2182** | 0.4362 | 0.5727 | 0.2105 | 0.6623 |
| GPT-4o (Hurst et al., 2024) | 0.1818 | **0.5667** | 0.6300 | 0.1711 | 0.6525 |
| LLaVA-OneVision (Li et al., 2024) | 0.1455 | 0.4810 | 0.4537 | 0.1842 | 0.5869 |
| Qwen2-VL (Wang et al., 2024b) | 0.1091 | 0.5000 | 0.5991 | 0.1974 | 0.5967 |
| GPT-5-mini (OpenAI, 2025) | 0.0909 | 0.3095 | 0.5991 | 0.1600 | 0.7180 |
| GeoScout-Agent | 0.1636 | 0.4459 | **0.6388** | **0.2500** | **0.7705** |

Despite these gains, absolute performance remains low for crop classification and farm pond change detection, indicating persistent challenges in capturing long-term temporal dependencies. These results suggest that

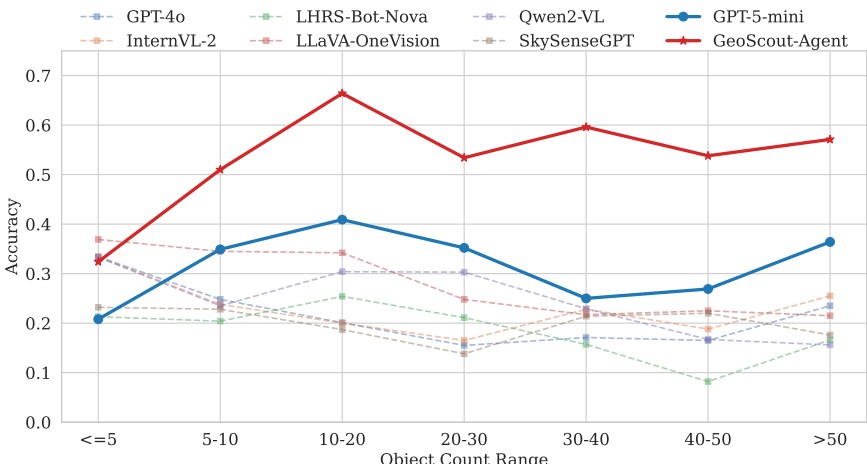

Figure 5: Object Density vs. Counting Accuracy.

current VLM-based approaches are insufficient for complex temporal reasoning and motivate future work on stronger temporal modeling mechanisms.

## 4.2 Accuracy Latency Tradeoff in Tool-Augmented Geospatial Reasoning

While *GeoScout-Agent* demonstrates substantial accuracy gains over single-pass inference, these improvements come with a measurable increase in inference latency and computational overhead. Tool-augmented inference requires iterative perception, state accumulation, and external module invocation, which collectively extend end-to-end response time. Empirically, *GeoScout-Agent* operates at an average latency of 4.30 seconds per sample, compared to 0.90 seconds per sample for the tool-free baseline. A complementary trend is observed in token consumption: tool-augmented inference uses an average of $1,884$ total tokens per interaction, compared to 779 tokens without tools, resulting in a $2.42\times$ increase. This growth is primarily driven by persistent agent state tracking, intermediate tool outputs, and iterative reasoning traces. Tool invocation statistics further indicate that *GeoScout-Agent* performs a mean of 2.79 tool calls per sample, highlighting that performance gains arise from selective, rather than exhaustive, tool usage.

From a deployment perspective, this trade-off is most appropriate for high-value EO workflows in which accuracy on difficult samples matters more than raw throughput. Examples include disaster assessment, infrastructure monitoring, environmental auditing, and humanitarian mapping, where small counting or classification errors can materially affect downstream decisions and manual verification is already costly. In such settings, a tool-augmented agent can serve as a decision-support system that selectively expends additional computation on ambiguous images rather than replacing fast screening models in all cases. Conversely, for large-scale bulk inference over millions of images, the current framework would likely be too expensive without further optimization, caching, or routing policies. We therefore view GeoScout-Agent as most immediately applicable in accuracy-critical, analyst-in-the-loop EO pipelines.

## 4.3 Generality Across Base Models

To demonstrate the generality of our framework, we have conducted additional experiments using Qwen3-VL-30B-A3B-Instruct as the base orchestrator (Figure 4). The agentic framework yields a consistent overall improvement from 47.16% to 56.63% when applied to Qwen3-VL, with particularly large gains in counting tasks (e.g., +66.8% relative in general aircraft counting, +73.1% in general vehicle counting, +49.3% in trees counting). This confirms that the performance benefits of our iterative perception-action loop generalize across different base models and are not specific to GPT-5-mini.

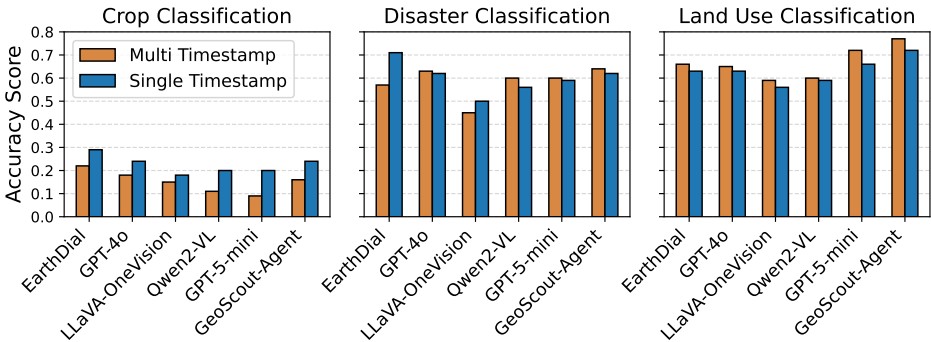

Figure 6: Single- versus multi-temporal performance comparison across crop classification, disaster classification, and land use classification tasks.

### 4.4 Qualitative Case Study

We present a representative qualitative example involving dense vehicle counting in satellite imagery. As shown in Fig. 2, the tool-free model produces a coherent explanation but miscounts due to occlusion and tightly packed objects. In contrast, Fig. 3 illustrates how tool use enables region refinement, instance separation, and verification through zoomed inspection, with *GeoScout-Agent* producing the correct count and demonstrating how multi-step perception and targeted visual analysis address failure modes of single-pass inference.

## 5 Ablation and Diagnostic Analysis

**Counting Accuracy as a Function of Object Density:** Figure 5 analyzes counting accuracy across object density regimes, from sparse scenes ($\leq 5$ objects) to highly dense scenes ($> 50$ objects). Most baseline models exhibit sharp performance degradation as density increases, particularly beyond 20 objects, reflecting challenges due to occlusion, overlap, and instance ambiguity under single-pass inference. Although GPT-5-mini is more stable than other baselines, its accuracy still declines in high-density settings.

In contrast, *GeoScout-Agent* achieves consistently higher accuracy across all density ranges, with the largest gains in medium and high-density regimes. This robustness demonstrates that iterative zooming, instance separation, and verification enable reliable counting as scene complexity increases, directly mitigating a core limitation of monolithic VLMs.

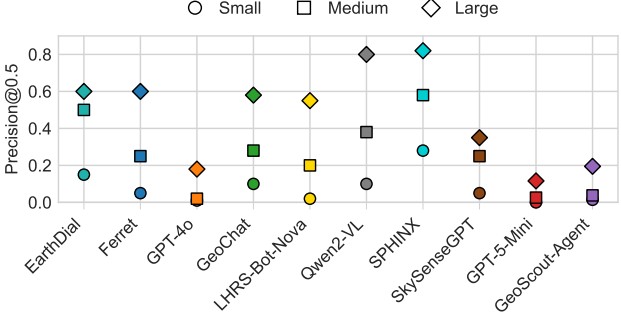

Figure 7: Performance comparison of different models on Referring Expression Detection across various object sizes

**Single vs. Multi-Temporal Data:** Figure 6 compares single- and multi-temporal performance for crop, disaster, and land use classification. Multi-temporal inputs reduce accuracy for crop classification across most models, including GPT-5-mini, indicating that temporal variability introduces noise without explicit

alignment. In contrast, disaster classification benefits from pre- and post-event observations, while land use classification shows consistent gains due to temporal stability. When combined, multi-temporal inputs and *GeoScout-Agent* yield compounding improvements for temporally informative tasks.

**Effect of Object Size on Referring Expression Detection:** Figure 7 presents referring expression detection performance across object sizes. GPT-5-mini performs poorly across all regimes, revealing limitations in fine-grained localization. Tool augmentation offers only marginal gains, particularly for large objects, suggesting that failures primarily arise from representational and grounding limitations rather than insufficient visual resolution.

**High-Resolution Zoom Baseline:**

To verify that these gains do not arise simply from exposing the model to more pixels, we additionally compare against a passive high-resolution baseline in which GPT-5-mini receives all inputs at 4× interpolated resolution, but without any iterative tool use. This baseline produces virtually no overall improvement over the standard single-pass setting (0.520 vs. 0.521), whereas *GeoScout-Agent* reaches 0.610. The result indicates that the benefit does not come from higher resolution alone, but from adaptive evidence gathering: the agent decides when to zoom, where to inspect, and how to use intermediate observations to refine its prediction. Full task-wise results are reported in Table 7 in Appendix A.

**Baseline Performance of Individual Tools:** To assess the contribution of the remaining tools beyond zoom, we evaluate DINOv3, SAM3, and code execution in isolation, without coordinated multi-step reasoning. As standalone models, these tools struggle significantly with the fine-grained localization and dense clutter typical of Earth observation imagery. As illustrated in the failure cases (Figure 8), DINOv3's class-agnostic clustering yields poor spatial precision (mean IoU of 8.5%), failing to distinguish small, tightly packed objects from visually similar background regions. Similarly, while SAM3 possesses strong zero-shot capabilities, its application to unguided counting tasks achieves only 29.32% accuracy. As shown in the failure cases (Figure 9), single-step segmentation frequently merges adjacent instances, falsely segments object shadows as separate entities, and spuriously classifies background regions.

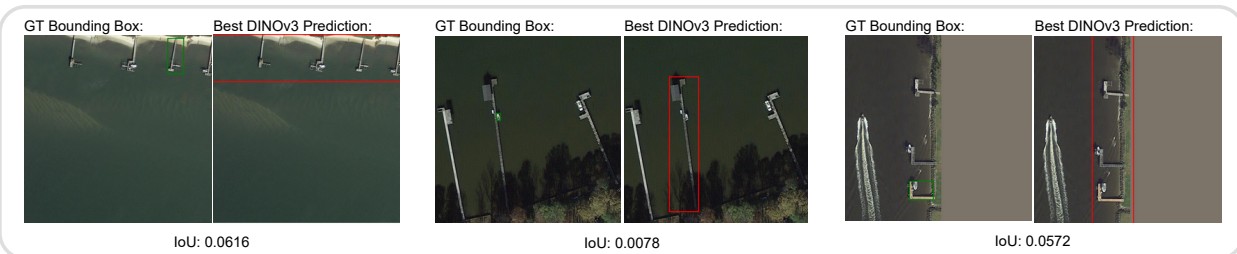

Figure 8: DINOv3 Failure Cases on GEO-bench Ref-Det. Representative low-scoring examples from unsupervised object detection using DINOv3 feature clustering. Green boxes indicate ground truth annotations. DINOv3 produces multiple candidate bounding boxes per image; for clarity, only the predicted box with the highest IoU to the ground truth (i.e., the best-matched prediction) is shown in red. Despite selecting the best match, the low IoU scores (0.0078-0.0616) demonstrate the model's difficulty with small object localization and fine-grained spatial reasoning in remote sensing imagery.

The code-execution baseline further underscores the limitations of isolated approaches, achieving an overall accuracy of just 7.97%. This demonstrates that deterministic, pixel-level analysis without iterative visual grounding is fundamentally insufficient for complex geospatial tasks. Ultimately, these baselines confirm that the performance gains observed in *GeoScout-Agent* stem from integrating complementary tools rather than the absolute strength of any single component. Within our framework, DINOv3 provides a coarse discovery signal, SAM3 enables targeted region verification via language model cross-checking, and code execution handles geometric transformations and pixel statistics. Complete implementation details, per-task quantitative breakdowns, and qualitative failure cases for all baselines are provided in Appendix A.

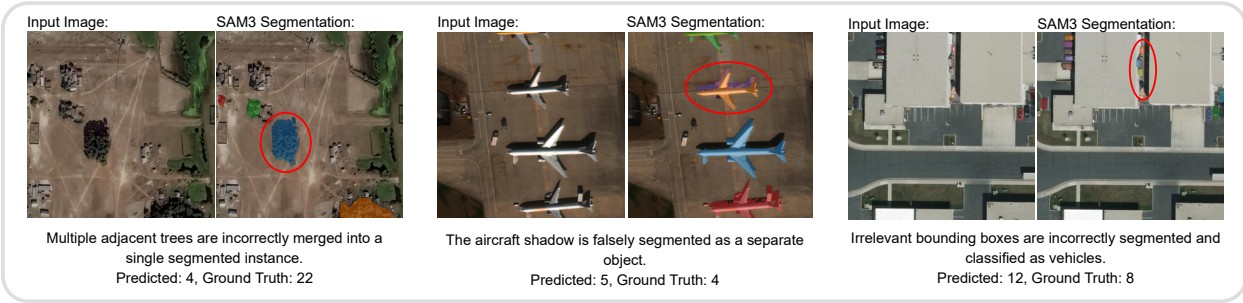

Figure 9: Representative failure cases of the SAM3 baseline on Counting Tasks, illustrating merged instances, shadow-induced false positives, and spurious object segmentation.

# 6 Conclusion

Earth observation requires reasoning systems that extend beyond passive visual recognition to support iterative, verifiable analysis of complex geospatial scenes, as many failures on GeoBench-VLM stem from the limitations of single-pass inference rather than model capacity alone. We introduce *GeoScout-Agent*, a tool-augmented agent built on GPT-5-mini that enables active perception through iterative inspection, local refinement, and explicit verification using segmentation, detection, zooming, sharpening, and lightweight code execution. Across GeoBench-VLM tasks, this agentic formulation consistently improves performance, with the largest gains observed in object counting, scene understanding, object classification, and temporal analysis, while fine-grained localization and referring expression grounding remain challenging. Overall, our findings demonstrate that explicit tool use is a principled and effective mechanism for improving geospatial reasoning over single-pass VLM inference, while highlighting the need for future work on stronger temporal modeling and spatial grounding in Earth observation.

# 7 Limitations

The proposed framework operates under practical resource constraints that influence model selection. We deliberately exclude open-source VLMs that require enterprise-scale GPUs, as well as high-cost proprietary frontier models, since the iterative nature of tool-augmented inference incurs substantial token and compute overhead. Instead, our evaluation emphasizes cost-efficient reasoning architectures that better reflect realistic deployment settings. While this choice improves accessibility and reproducibility, it may limit absolute performance ceilings relative to more resource-intensive alternatives. Additionally, due to computational budget constraints, all experiments are conducted using a single evaluation run per configuration. We do not perform repeated runs or extensive ablation studies isolating individual tools. Consequently, our analysis focuses on comparative trends rather than variability across runs.

# 8 Future Work

Future work could extend evaluations beyond final answer correctness to assess the quality of intermediate reasoning steps. Leveraging recent agentic benchmarks, such as ThinkGeo(Shabbir et al., 2025), GTA(Wang et al., 2024a), and GAIA(Mialon et al., 2023), to measure fine-grained metrics like tool selection accuracy and argument formatting, allowing for a more precise diagnosis of planning versus perception failures. Furthermore, future research should also strengthen temporal modeling for dynamic scenes and improve fine-grained spatial grounding through multi-scale or native-resolution architectures to better handle small objects and dense regions.

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

# A    Baseline Performance of Individual Tools

To analyze the contribution of individual tools, we report baselines with *GeoScout-Agent*'s tools enabled in isolation. We evaluate vision-only, segmentation-only, and code-execution settings, each without coordinated tool use or multi-step reasoning. These baselines reveal the limitations of single-module approaches and show that performance gains in the main results stem from integrating complementary tools rather than any individual component.

## A.1    SAM3 Baseline

SAM3[1] is a foundation segmentation model that produces high-quality, zero-shot object masks across diverse visual domains without task-specific training.

To compute the SAM3 baseline, we apply SAM3 to all counting tasks in the single MCQ subset. For each task, the object category is extracted from the question prompt and provided as text input to SAM3 (e.g., "How many pickup trucks are there in the image?" → *pickup trucks*). SAM3 produces segmentation masks for the queried category, and the total number of segmented instances is taken as the predicted count. For multiple-choice evaluation, the option closest to this predicted count is selected; if two options are equally distant, one is chosen at random. The resulting per-task and overall performance is summarized in Table 4.

Table 4: SAM3 baseline performance on counting tasks.

| Task | Correct | Total | Accuracy (%) |
|---|---|---|---|
| Building Counting | 44 | 170 | 25.88 |
| General Aircraft Counting | 55 | 100 | 55.00 |
| General Vehicle Counting | 81 | 150 | 54.00 |
| Marine Debris Counting | 17 | 100 | 17.00 |
| Specific Aircraft Type Counting | 38 | 140 | 27.14 |
| Specific Vehicle Type Counting | 23 | 224 | 10.27 |
| Trees Counting | 8 | 85 | 9.41 |
| Water Bodies Counting | 43 | 85 | 50.59 |
| Overall | 309 | 1054 | 29.32 |

While SAM3 demonstrates strong zero-shot segmentation capabilities in general vision tasks, its standalone application to remote sensing counting tasks reveals critical limitations. Single-step segmentation without iterative verification suffers from three recurring failure modes: adjacent instances being merged into a single mask, object shadows being falsely segmented as separate instances, and irrelevant background regions being spuriously classified as target objects. These errors compound in dense and cluttered satellite scenes, resulting in an overall counting accuracy of only 29.32% (Table 4). This confirms that segmentation quality alone is insufficient for reliable object counting in Earth observation, and that SAM3 contributes meaningfully to *GeoScout-Agent* only when invoked within the broader iterative reasoning loop where the language model can cross-check and verify its outputs.

## A.2    DINOv3 Baseline

DINOv3 provides unsupervised, class-agnostic object detection by leveraging self-supervised vision transformer representations. The module extracts dense patch-level features from a DINOv3 checkpoint pretrained on satellite imagery (DINOv3-ViT-L/16, 493M parameters[2]).

Images are resized so that the shortest edge is 518 pixels before passing through the backbone, and the resulting patch tokens are reshaped into a spatial grid of feature vectors (patch size of $14 \times 14$ pixels per patch). We perform unsupervised region discovery by $L_2$-normalizing these patch features and clustering them with K-means (fixed $K = 4$ clusters). The background cluster is identified heuristically as the cluster containing the largest number of patches, under the assumption that background regions dominate the image

---

[1] https://huggingface.co/facebook/sam3
[2] https://huggingface.co/facebook/DINOv3-vitl16-pretrain-sat493m

spatially. The remaining $K-1$ clusters are treated as candidate object regions, we generate binary masks, upsample them to the original image resolution, extract contours, and compute bounding rectangles.

For referring expression detection, evaluation is performed using standard intersection-over-union (IoU) based metrics. The detector produces multiple candidate bounding boxes per image. For each ground-truth box, IoU is computed against all predicted boxes, and the maximum IoU is retained as the score for that ground truth. Mean IoU is then reported as the average of these per-ground-truth maximum IoUs. Using this protocol, the DINOv3-based baseline achieves a mean IoU of 8.5%.

Although DINOv3 provides robust unsupervised feature representations, its standalone deployment as an object detector struggles significantly with the fine-grained localization demands of remote sensing imagery. The K-means clustering approach produces bounding boxes with very low IoU scores ranging from 0.0078 to 0.0616, even when selecting the best-matched prediction per ground truth. This poor spatial precision stems from DINOv3's class-agnostic clustering being ill-suited to distinguishing small, densely packed objects from visually similar background regions at satellite resolution. With a mean IoU of only 8.5% and precision at IoU threshold 0.5 of just 4.37% (Table 5), DINOv3 alone is an unreliable localization tool. However, within *GeoScout-Agent* its value lies not in precise localization but in providing a coarse discovery signal that guides subsequent tool invocations such as targeted zooming and SAM3 segmentation toward regions of interest.

Table 5: DINOv3 baseline performance on referring expression detection.

| Metric | Prec@0.25 | Prec@0.50 | Prec@0.75 |
|---|---|---|---|
| **Precision (%)** | 9.62 | 4.37 | 0.15 |

## A.3 Code Execution Tool Baseline

For the code-execution baseline, GPT-5-mini is prompted to generate deterministic Python code for each of the eight counting tasks: Building, General Aircraft, General Vehicle, Marine Debris, Specific Aircraft Type, Specific Vehicle Type, Trees, and Water Bodies. The prompt template instructs the model to write code that processes a PIL Image object and returns an integer count. If the generated code raises an exception during execution, the model is re-prompted with the error traceback appended to the original prompt. Each counting task is evaluated independently using a task-specific prompt. The generated code is executed in a constrained environment, and the predicted integer is mapped to a multiple-choice answer by selecting the closest option; if two options are equidistant, one is chosen at random.

As shown in Table 6, this baseline achieves an overall accuracy of only 7.97%, with per-task accuracy ranging from 3.57% to 21.00%. This poor performance demonstrates that deterministic pixel-level analysis without visual grounding is fundamentally insufficient for object counting in complex satellite imagery. Within *GeoScout-Agent*, code execution therefore serves a complementary role-handling geometric transformations and pixel statistics-rather than acting as a primary counting mechanism.

Table 6: Code execution baseline performance on counting tasks.

| Task | Correct | Total | Accuracy (%) |
|---|---|---|---|
| Building Counting | 13 | 170 | 7.65 |
| General Aircraft Counting | 6 | 100 | 6.00 |
| General Vehicle Counting | 16 | 150 | 10.67 |
| Marine Debris Counting | 21 | 100 | 21.00 |
| Specific Aircraft Type Counting | 5 | 140 | 3.57 |
| Specific Vehicle Type Counting | 11 | 224 | 4.91 |
| Trees Counting | 4 | 85 | 4.71 |
| Water Bodies Counting | 8 | 85 | 9.41 |
| Overall | 84 | 1054 | 7.97 |

**Prompting Details**

All code-execution prompts follow the same structure, differing only in the target object category and task-specific query examples:

---

**Code Execution Prompt Template (Tool Baseline)**

You are an expert computer vision engineer.
Task: Given an input image containing {OBJECT}, write Python code that counts the total number of {OBJECT} present in the image.
The code should be able to answer questions like: {TASK-SPECIFIC QUERY VARIANTS}
Constraints:
- The image is already loaded as a PIL Image object named 'image'.
- You may use: PIL, NumPy, OpenCV and other standard Python libraries.
- The function must be deterministic.
- The output must be a single integer representing the number of buildings/structures in the image.
Output requirements of code:
- Return ONLY the integer count.
- No print statements.
- No explanations.
- No comments outside the code.
- Output only valid Python code.

---

In this template, `OBJECT` denotes the target object category specific to each counting task (e.g., trees, water bodies, vehicles), and `TASK-SPECIFIC-QUERY-VARIANTS` comprises all unique question formulations from the single-answer subset that pertain to the given task, providing the model with representative examples of expected queries.

### A.4  Zoom Tool Baseline

A potential confounding factor in our results is whether the performance gains of *GeoScout-Agent* stem from the agentic framework or simply from the increased spatial resolution provided by the zooming tool. To isolate these effects, we evaluate a High-Resolution (HR) Baseline, where the base GPT-5-mini model receives all images at $4\times$ the standard pixel resolution, providing the model with spatial detail equivalent to that accessible through the zoom tool, but without iterative tool use or feedback. Images were upsampled using LANCZOS interpolation to maintain edge sharpness during rescaling

As shown in Table 7, the HR Baseline yields a negligible change in average accuracy (moving from 0.520 to 0.521), whereas *GeoScout-Agent* achieves 0.610 (+9% absolute improvement). These results demonstrate that the benefit of our framework is not a mere function of pixel count. Instead, the value of the zoom tool lies in its adaptive, context-driven application: the LLM orchestrator dynamically selects the zoom center and scale based on intermediate observations. This iterative strategy allows the model to resolve ambiguities through active perception, a capability that is fundamentally absent in fixed high-resolution inference.

## B  Hyperparameters Settings and Implementation Details

All experiments were conducted on a single NVIDIA A6000 GPU. The hyperparameters listed in Table 8 are fixed and shared across all tasks.

## C  Image Processing Tools

All image operations are implemented using the Pillow library.[3]

---

[3]https://pillow.readthedocs.io

Table 7: Ablation study comparing the base model, a static high-resolution baseline (4× interpolation), and the iterative *GeoScout-Agent* across GeoBench-VLM task categories.

| Task Category | GPT-5-mini-baseline | GPT-5-mini-high-res | GeoScout-Agent |
|---|---|---|---|
| Aircraft Classification | 0.780 | 0.808 | **0.820** |
| Building Counting | 0.259 | 0.235 | **0.435** |
| Crop Classification | 0.200 | 0.218 | **0.236** |
| Disaster Classification | 0.590 | 0.582 | **0.617** |
| Fire Risk Assessment | 0.208 | 0.158 | **0.250** |
| General Aircraft Counting | 0.470 | 0.440 | **0.550** |
| General Vehicle Counting | 0.280 | 0.353 | **0.500** |
| Land Use Classification | 0.656 | 0.649 | **0.717** |
| Marine Debris Counting | 0.350 | 0.260 | **0.450** |
| Scene Classification | 0.872 | 0.874 | **0.899** |
| Ship Classification | 0.604 | 0.631 | **0.691** |
| Specific Aircraft Counting | 0.171 | 0.243 | **0.297** |
| Specific Vehicle Counting | 0.241 | 0.210 | **0.368** |
| Spatial Relationship | 0.717 | 0.730 | **0.755** |
| Tree Health Assessment | 0.229 | 0.217 | **0.372** |
| Trees Counting | 0.259 | 0.318 | **0.577** |
| Water Bodies Counting | 0.329 | 0.329 | **0.588** |
| **OVERALL** | 0.520 | 0.521 | **0.610** |

Table 8: Hyperparameters used across all experiments.

| Component | Parameter | Value |
|---|---|---|
| Language Model | Model | gpt-5-mini |
| Language Model | Temperature | 0.3 |
| SAM3 Segmentation | Threshold | 0.5 |
| DINOv3 Detection | Feature dim | 518 |
| DINOv3 Detection | Patch size | 14 |
| DINOv3 Detection | #Clusters | 4 |
| DINOv3 Detection | Box threshold | 0.02 |
| DINOv3 Detection | Random state | 42 |
| Image Tools | Sharpen intensity Default | 2.0 |
| Image Tools | Sharpen radius Default | 2 |
| Image Tools | Sharpen threshold Default | 3 |
| Image Tools | Zoom factor Default | 2.0 |
| Image Tools | Output size | 448 |

## C.1 Image Sharpening Tool

The sharpening tool applies an unsharp mask filter to enhance blurred or low-quality images, revealing fine details such as text or object boundaries. The intensity parameter controls sharpening strength, where higher values produce more pronounced edge enhancement.

```
def sharpen_image_tool(image_path: str, intensity: float = 2.0) -> str:
    img = Image.open(image_path)

    sharpened_img = img.filter(ImageFilter.UnsharpMask(
        radius=2,
        percent=int(intensity * 100),
        threshold=3
    ))
```

```
    output_path = f"{base}_sharpened_{intensity}{ext}"

    sharpened_img.save(output_path)
    return output_path
```

### C.2 Image Zoom Tool

The zoom tool extracts and magnifies a region of interest specified by normalized coordinates $(x, y) \in [0, 1]^2$, where $(0.5, 0.5)$ represents the image center. The cropped region is resized to $448 \times 448$ pixels using Lanczos interpolation.

```
def zoom_image_tool(image_path: str, x: float, y: float, zoom_factor: float = 2.0) -> str:
    img = Image.open(image_path)
    w, h = img.size

    # Convert normalized coordinates to pixels
    center_x = int(x * w) if x <= 1.0 else int(x)
    center_y = int(y * h) if y <= 1.0 else int(y)

    # Calculate crop dimensions
    crop_w = max(1, int(w / zoom_factor))
    crop_h = max(1, int(h / zoom_factor))

    # Define crop box
    left = max(0, center_x - crop_w // 2)
    top = max(0, center_y - crop_h // 2)
    right = min(w, left + crop_w)
    bottom = min(h, top + crop_h)

    # Crop and resize
    cropped = img.crop((left, top, right, bottom))
    zoomed_img = cropped.resize((448, 448), Image.LANCZOS)

    output_path = f"{base}_zoomed_{zoom_factor}_{x}_{y}{ext}"
    zoomed_img.save(output_path)
    return output_path
```

## D   Prompt Templates

We provide the complete prompt templates used in our evaluation framework. All prompts were designed to elicit structured reasoning and consistent answer formatting across experimental conditions.

### D.1   Baseline System Prompt

The following system prompt defines the tool-augmented VQA agent's capabilities and instructions for multi-turn reasoning.

**Single MCQ Task**

You are an expert Visual Question Answering assistant. Analyze images carefully and provide accurate answers based on what you can see.
Question: {*question*} Options: {*options_text*}
Please analyze the provided image carefully. Follow these steps:
1. First, explain your reasoning about what you observe in the image

2. Then, select the best option from the list above
3. Format your response as: Reason: [Your detailed analysis and reasoning] Answer: [Single letter A, B, C, D, or E]

**Captioning Task**

You are an expert image captioning assistant specializing in aerial and satellite imagery analysis. Task: {*prompt*} Analyze the provided image and generate a detailed caption.
Focus on:
- Main objects and structures visible
- Spatial relationships between elements
- Colors, sizes, and notable features
- Overall scene context and setting
Provide a comprehensive caption that captures all important details visible in the image.

**Referring Expression Detection Task**

You are an expert at detecting and localizing objects in satellite/aerial images. Given a referring expression, output the bounding box coordinates of the described object(s).
Output ONLY coordinates in [[x1, y1, x2, y2], ...] format.
Referring Expression: {*prompt_text*}
Image size: {*image_size*} × {*image_size*} pixels
IMPORTANT: The number at the beginning of the referring expression indicates EXACTLY how many objects to detect. In this case, you must output EXACTLY {*num_objects*} bounding box(es).
Locate the object(s) described by the referring expression in the image.
Output ONLY the bounding box coordinates in this exact format: [[x1, y1, x2, y2]]
Where x1,y1 is top-left corner and x2,y2 is bottom-right corner.
For multiple objects, use: [[x1, y1, x2, y2], [x1, y1, x2, y2], ...]
Coordinates must be integers in pixel values (0 to {*image_size*}). Output EXACTLY {*num_objects*} box(es), nothing more, nothing less.

**Temporal Task**

You are an expert Visual Question Answering assistant specializing in analyzing temporal changes in satellite/aerial imagery. Analyze images carefully and provide accurate answers based on what you can see. You are provided with {*num_images*} images showing temporal changes.
Question:{*question*} Options: {*options_text*}
Please analyze the provided images carefully, comparing the temporal changes. Follow these steps:
1. First, explain your reasoning about what you observe across the images
2. Then, select the best option from the list above
3. Format your response as: Reason: [Your detailed analysis and reasoning] Answer: [Single letter A, B, C, D, or E]

## D.2 Agent System Prompt

The following system prompt defines the tool-augmented VQA agent's capabilities and instructions for multi-turn reasoning.

**Main System Prompt (Agent with Tools)**

You are an expert Visual Question Answering (VQA) assistant with access to powerful image analysis and web search tools. Your goal is to provide accurate answers to questions about images.
**AVAILABLE TOOLS AND HOW TO USE THEM:**
1. **sharpen_image_tool(image_path: str, intensity: float = 2.0) -> str**
- Purpose: Enhance blurred or low-quality images to reveal fine details
- Input: image_path (file path), intensity (1.0-3.0, higher = more sharpening)
- Output: Path to the sharpened image
- When to use: Image is blurry, text is hard to read, or fine details are unclear

- Performance: Improves clarity of edges, text, and small features
2. **zoom_image_tool(image_path: str, x: float, y: float, zoom_factor: float = 2.0) -> str**
- Purpose: Magnify a specific region of an image for detailed inspection
- Input: image_path, x/y coordinates (0.0-1.0 as ratio of width/height), zoom_factor
- Output: Path to the zoomed/cropped image
- When to use: Need to examine small objects, read distant text, or analyze a specific area
- Performance: Helps identify small objects, count items in dense areas, read fine text
3. **execut_code_tool(image_path: str, code_string: str) -> str**
- Purpose: Run ANY Python code to analyze OR manipulate images
- Input: image_path, Python code (variable 'image' is PIL Image object, output to 'result')
- Output: Path to the transformed/manipulated image
- When to use:
* ANALYSIS: Extract features, count pixels, measure properties, detect patterns
* MANIPULATION: Adjust brightness/contrast, apply filters, rotate, crop, color adjustments
- Performance: Extremely flexible - can do anything Python/PIL/OpenCV supports
- IMPORTANT: If you manipulate an image, the output path can be used in subsequent tool calls!
- Examples:
* Analysis: "import numpy as np; hist = np.histogram(np.array(image)); result = image"
* Manipulation: "from PIL import ImageEnhance; result = ImageEnhance.Contrast(image).enhance(2.0)"
* Custom filter: "result = image.filter (ImageFilter.EDGE_ENHANCE_MORE)"
4. **sam3(image_path: str, prompt: str) -> str**
- Purpose: Segment and highlight specific objects using text descriptions
- Input: image_path, prompt (object description like "cars", "buildings", "trees")
- Output: Path to image with colored segmentation masks overlaid
- When to use: Count objects, identify object locations, analyze object distribution
- Performance: Excellent for object counting, spatial analysis, and object identification
5. **DINOv3(image_path: str) -> str**
- Purpose: Automatically detect and box salient objects without labels
- Input: image_path
- Output: Path to image with bounding boxes around detected objects
- When to use: Discover unknown objects, find regions of interest, unsupervised object detection
- Performance: Good for finding distinct regions, objects, or structures without prior knowledge
**IMPORTANT INSTRUCTIONS:**
- ALWAYS analyze the question first to determine which tools might help
- Use tools EFFICIENTLY - you have a maximum of 3 tool calls, so choose wisely
- You can use MULTIPLE tools in sequence (e.g., sharpen → zoom → sam3), but be strategic
- For counting tasks, use sam3 or DINOv3
- For reading small text, use sharpen_image_tool or zoom_image_tool
- The conversation history shows all previous tool outputs - use them for your final answer
- After using tools, you MUST provide a clear, concise text answer based on ALL information gathered - If the image alone is sufficient, you may answer directly without tools
**MULTI-TURN STRATEGY:**
- First turn: Use necessary tools to gather information
- Subsequent turns: Analyze tool outputs and refine if needed
- Final turn: YOU MUST provide a TEXT response with your definitive answer
**CRITICAL: After using tools, you MUST provide a final text answer. Do not end the conversation with just tool calls - always conclude with your reasoning and answer in text format.**
Your responses should be accurate, well-reasoned, and based on the evidence from the image and tool outputs.

## Referring Expression Detection: Agent System Prompt

You are an expert Visual Question Answering (VQA) assistant specialized in detecting and localizing objects in satellite/aerial images based on referring expressions.
**YOUR TASK:**

Given an image and a referring expression (e.g., "1 baseball-diamond at the bottom"), you must identify and output the bounding box coordinates of the described object(s).
**OUTPUT FORMAT - CRITICAL:**
You MUST output your answer as bounding box coordinates in the following JSON format: [[x1, y1, x2, y2], [x1, y1, x2, y2], ...]
Where:
- x1, y1 = top-left corner coordinates
- x2, y2 = bottom-right corner coordinates
- Multiple boxes are separated by commas within the outer list
- Coordinates are in pixels (integer values)
**EXAMPLE OUTPUTS:**
- Single object: [[420, 556, 891, 1021]]
- Two objects: [[420, 556, 891, 1021], [100, 200, 300, 400]]
**AVAILABLE TOOLS AND WHEN TO USE THEM:**
1. **sharpen_image_tool** - Enhance blurry images to see details better
2. **zoom_image_tool** - Magnify specific regions for detailed inspection
3. **execute_code_tool** - Run custom Python code for image analysis
4. **sam3** - Segment objects using text prompts (useful for finding object boundaries)
5. **DINOv3** - Detect salient objects automatically (useful for discovering object locations)
**IMPORTANT INSTRUCTIONS:** - Analyze the image carefully to locate the described object(s)
- Pay attention to spatial references (top, bottom, left, right, center)
- Count the number of objects mentioned in the expression
- Provide coordinates that tightly bound the described object(s)
- Use tools if needed to better identify object locations
- Your final answer MUST be in the JSON bounding box format specified above
- Do NOT include any other text in your final answer - ONLY the coordinate list
**CRITICAL: Your final response must be ONLY the bounding box coordinates in the format [[x1, y1, x2, y2], ...]. No explanations, no reasoning - just the coordinates.**

The temporal and captioning tasks share the same agent system prompt as the primary agent, with no task-specific modifications.

### D.3 Agent Query Format

The agent prompt additionally includes the image file path for tool invocation and encourages strategic tool use.

**Single MCQ Task**

Image file path: {*img_path*}
Question: {*question*} Options: *options_text*
Analyze the provided image and answer the question by selecting the best option from the list above.
Consider using your available tools if they can help improve accuracy.
IMPORTANT: When calling tools, use the exact image path provided above: {*img_path*}
After gathering information (with or without tools), provide your final response in this format:
Reason: [Your detailed analysis and reasoning based on the image and any tool outputs] Answer: [Single letter A, B, C, D, or E]

**Captioning Task**

Image file path: {*image_path*}
Task: {*prompt*}
Analyze the provided image and generate a detailed caption. You may use your available tools to:
- Segment specific objects (sam3) to identify and count elements
- Detect objects automatically (DINOv3) to find structures
- Zoom into specific areas for detail

- Sharpen the image if needed
IMPORTANT: When calling tools, use the exact image path: {*image_path*}
After analyzing the image (with or without tools), provide your final caption that describes:
- Main objects and structures visible
- Spatial relationships between elements
- Notable features, colors, and sizes
- Overall scene context

### Referring Expression Detection Task

Image file path: {*image_path*}
Image size: {*image_size*} × {*image_size*} pixels
Referring Expression: {*prompt_text*}
IMPORTANT: The number at the beginning of the referring expression indicates EXACTLY how many objects to detect.
In this case, you must output EXACTLY {*num_objects*} bounding box(es).
Locate the object(s) described by the referring expression.
You may use available tools (sam3, DINOv3, zoom, etc.) to help identify the object location. IMPORTANT: When calling tools, use the exact image path: {*image_path*}
After analysis, output ONLY the bounding box coordinates in this exact format:
[[x1, y1, x2, y2]]
Where x1,y1 is top-left corner and x2,y2 is bottom-right corner.
For multiple objects: [[x1, y1, x2, y2], [x1, y1, x2, y2], ...]
Coordinates must be integers in pixel values (0 to {*image_size*}). Output EXACTLY {*num_objects*} box(es), nothing more, nothing less.

### Temporal Task

You are provided with {*num_images*} images showing temporal changes. Image file paths: {*image_paths_str*}
Question: {*question*} Options: {*options_text*}
Analyze the provided images (comparing temporal changes) and answer the question by selecting the best option from the list above.
Consider using your available tools if they can help improve accuracy.
IMPORTANT: When calling tools, use the exact image paths provided above.
After gathering information (with or without tools), provide your final response in this format:
Reason: [Your detailed analysis and reasoning based on the images and any tool outputs]
Answer: [Single letter A, B, C, D, or E]

### D.4 Agent Loop Termination Prompt

When the tool-call budget is exhausted, forcing reminders are applied to guarantee answer generation; no reminder is needed for captioning, and the temporal task follows the same reminder as the single-step MCQ task.

### Single MCQ Task

You have gathered sufficient information from tools. Now provide your final answer as ONLY a single letter (A, B, C, D, or E). Do NOT call any more tools.

### Referring Expression Detection Task

You have gathered sufficient information from tools. Now provide your final answer as ONLY the bounding box coordinates in the format [[x1, y1, x2, y2], ...]. Do NOT call any more tools. Do NOT include any explanations.

# E GeoBench-VLM Dataset Examples

We present illustrative examples from the GeoBench-VLM dataset (Fig 10). These samples highlight the diverse range of remote sensing tasks, categorized into Scene Understanding, Object Classification, Localization & Counting, Event Detection, Caption Generation, Temporal Understanding and Referring Expression Detection.

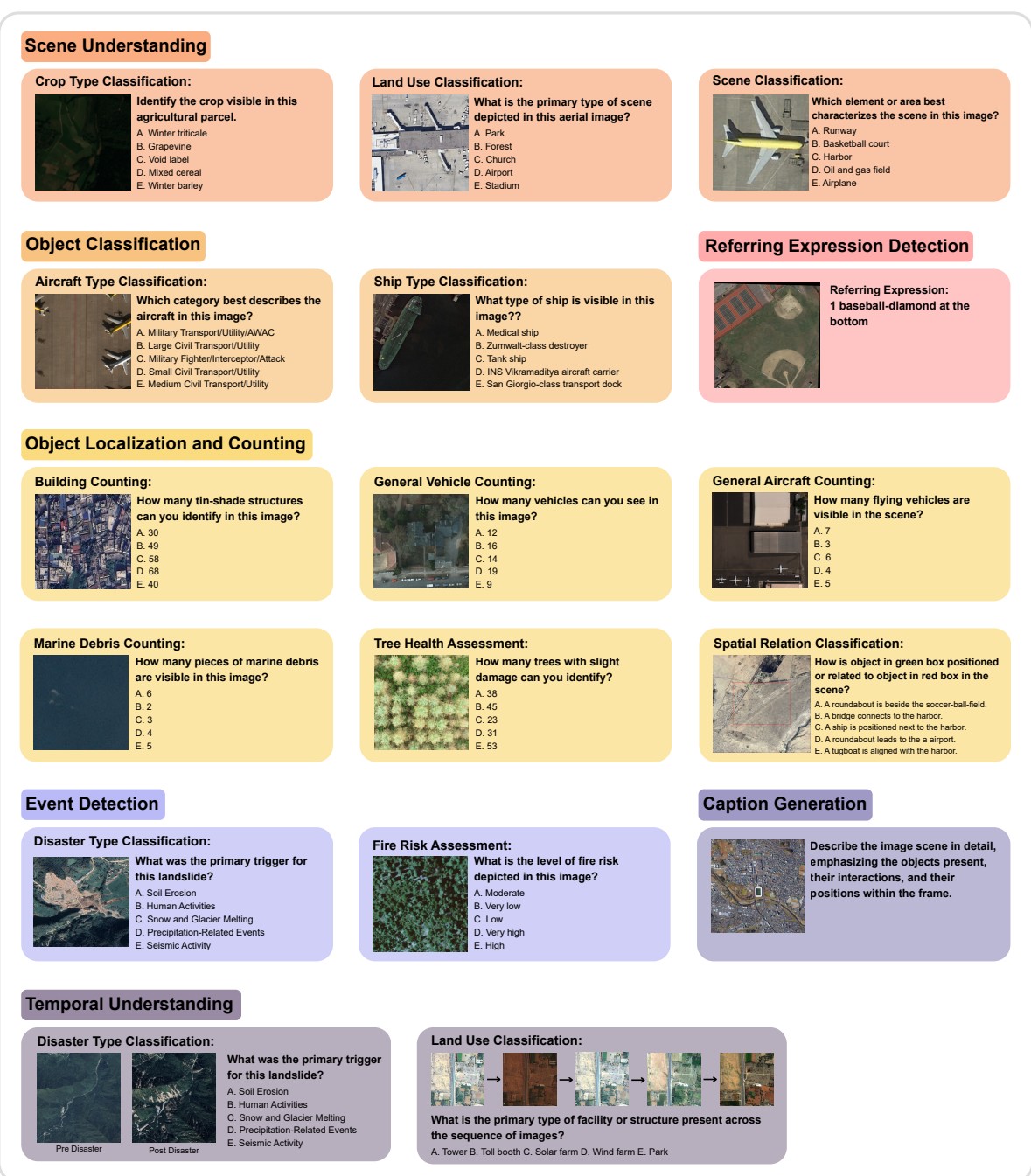

Figure 10: Qualitative examples from the GeoBench-VLM benchmark.

## F   Additional Qualitative Case Studies

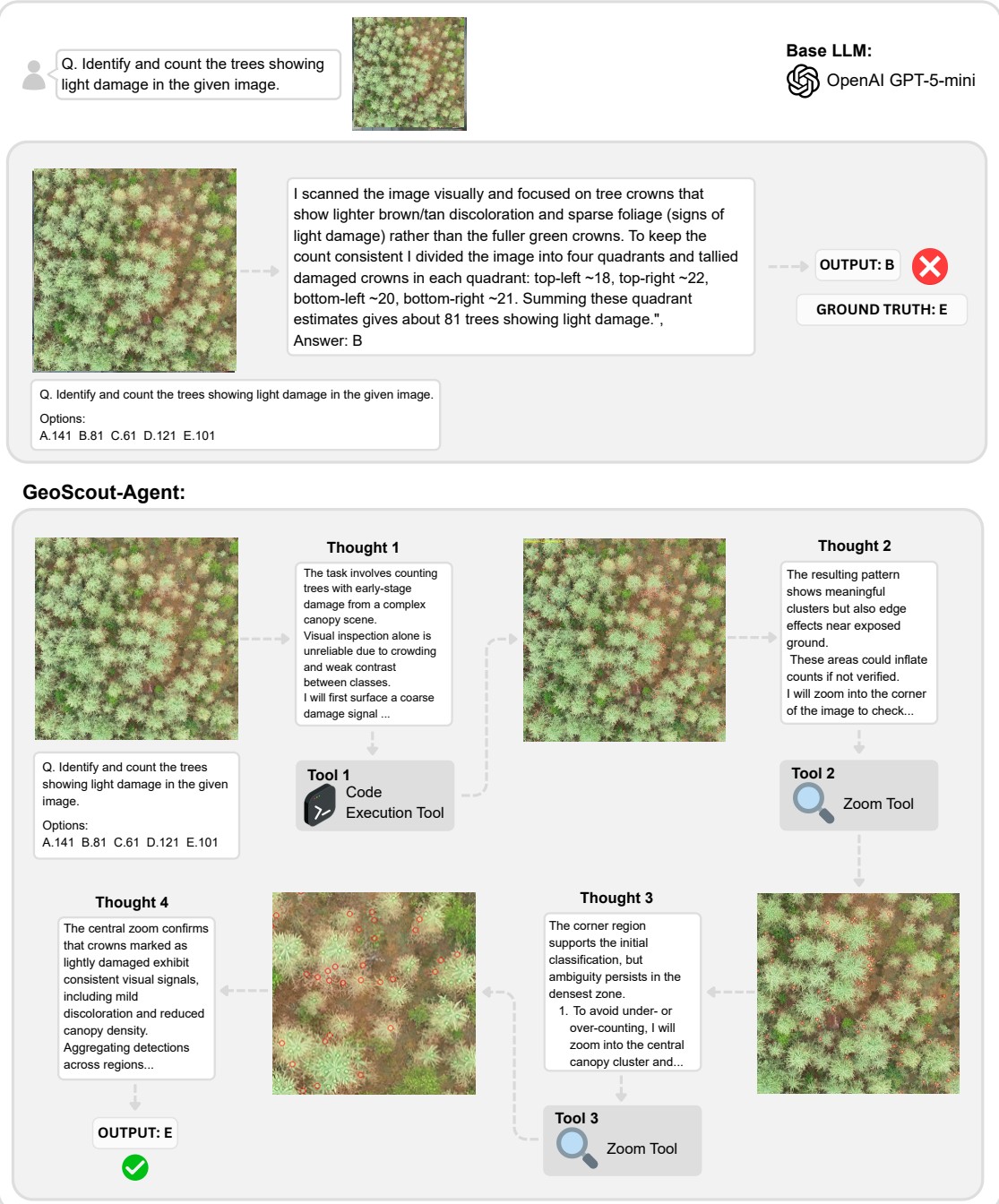

Figure 11: Qualitative case study on tree health assessment. GeoScout-Agent iteratively refines visual evidence through localized inspection, enabling more reliable discrimination of vegetation condition in the presence of clutter, shadows, and resolution variability.

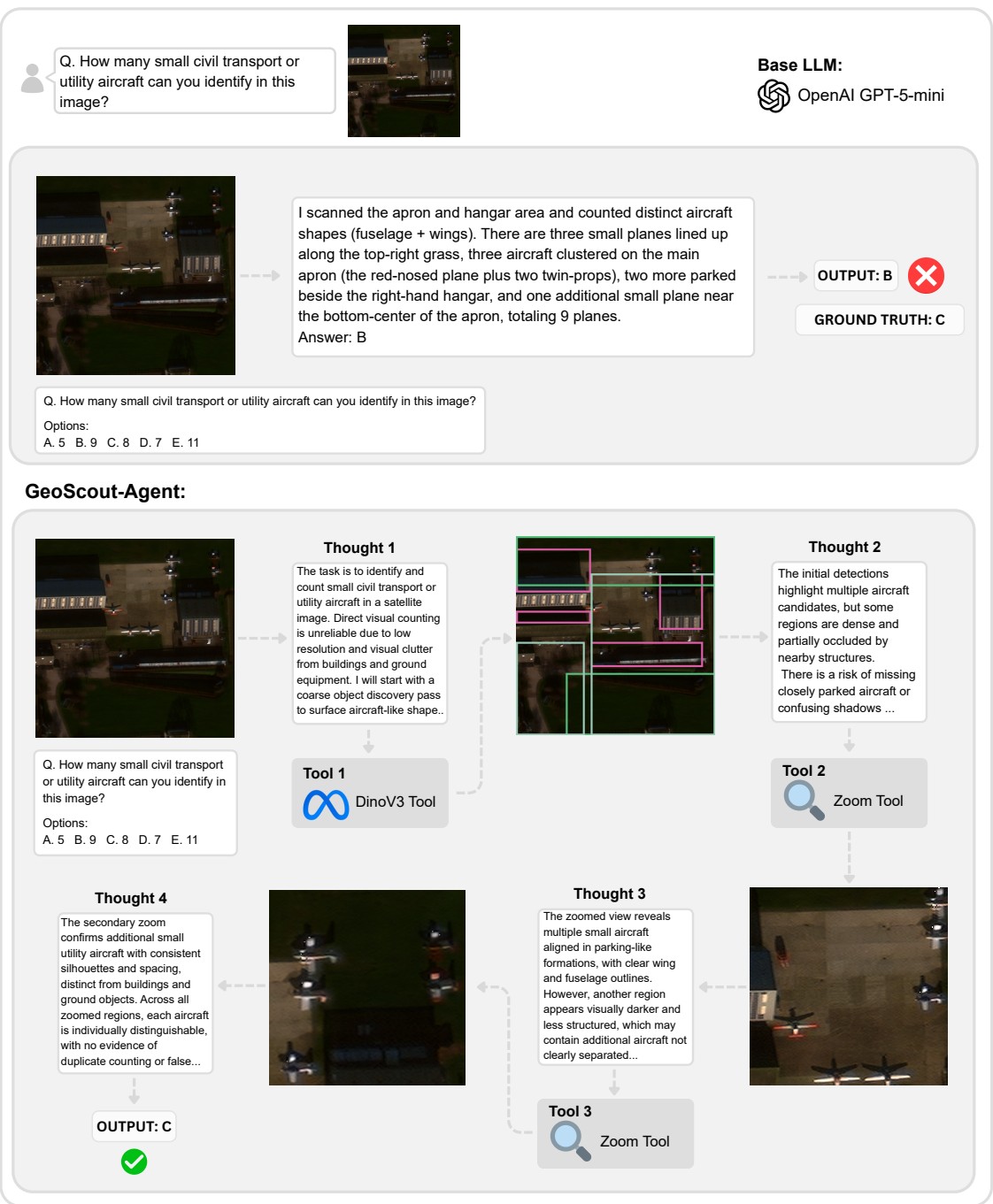

Figure 12: Qualitative case study on specific aircraft type counting. GeoScout-Agent resolves dense layouts and visually similar instances through multi-step object discovery, targeted zooming, and verification, illustrating how tool-assisted reasoning improves fine-grained counting accuracy.

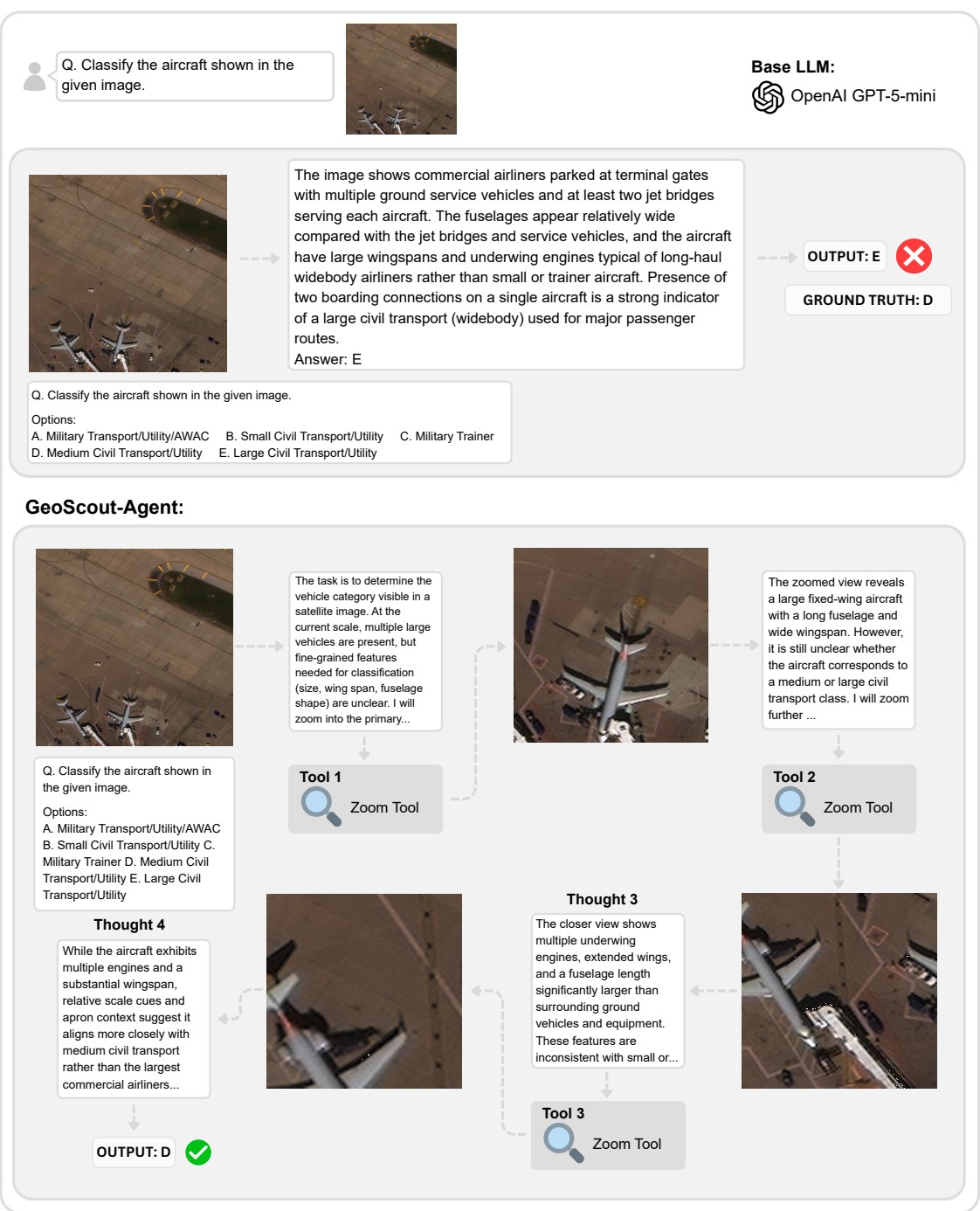

Figure 13: Qualitative case study on aircraft type classification. GeoScout-Agent leverages iterative visual refinement to isolate discriminative structural features, reducing ambiguity between closely related aircraft categories that commonly confound single-pass inference.

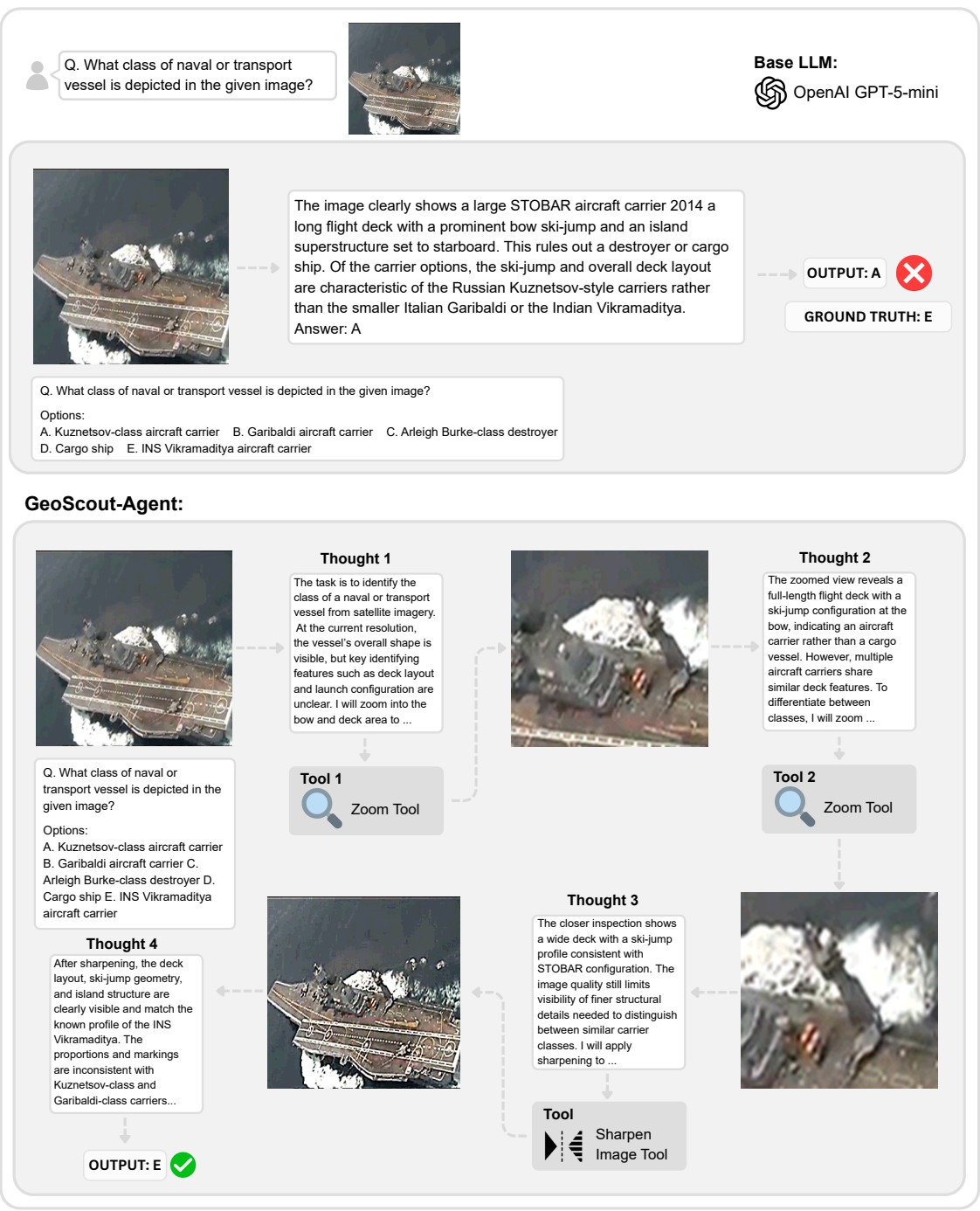

Figure 14: Qualitative case study on Ship Type Classification. GeoScout-Agent applies iterative zooming and sharpening to resolve fine-grained deck and superstructure cues, disambiguating closely related aircraft carrier classes that mislead single-pass LLM inference.

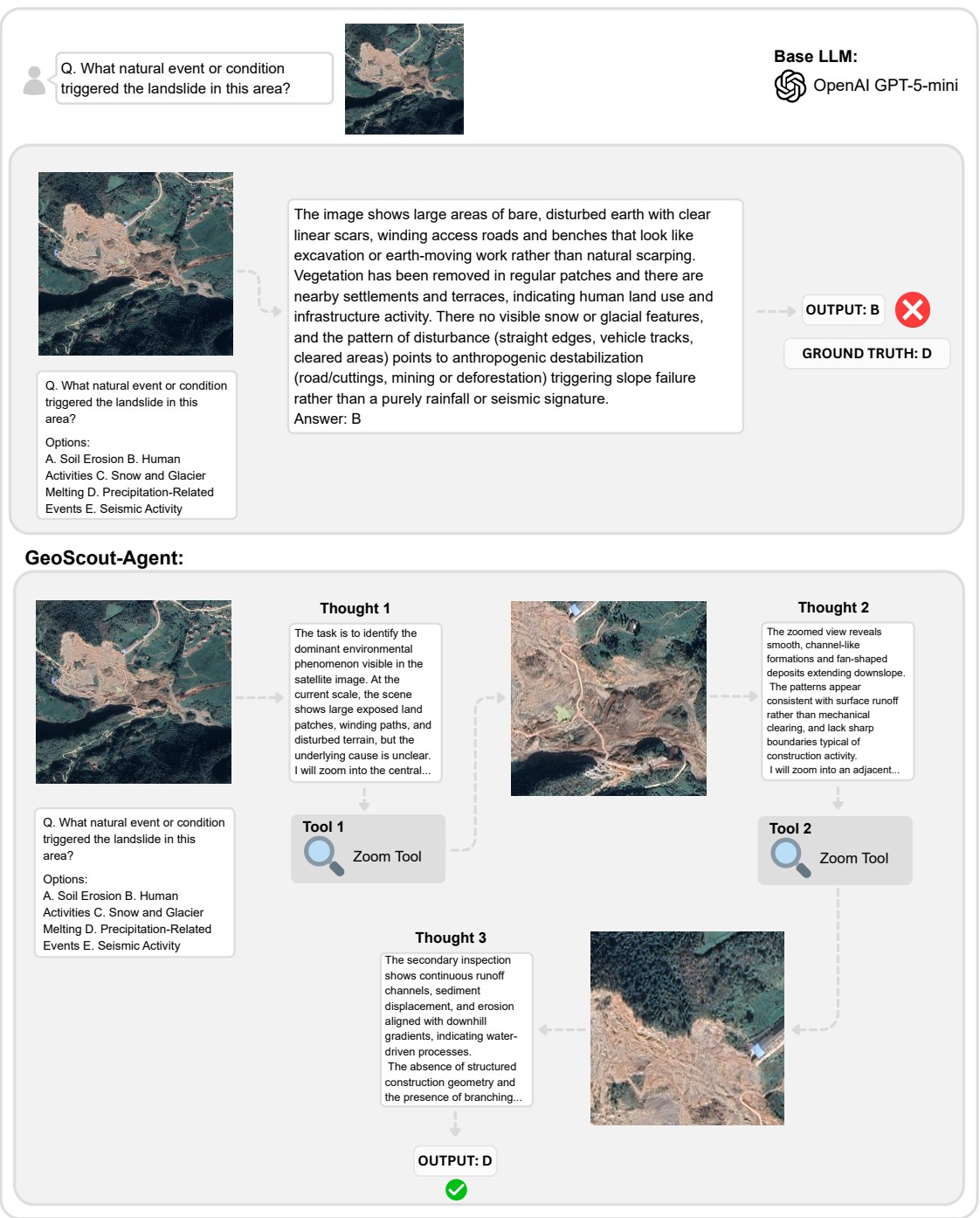

Figure 15: Qualitative case study on Disaster Type Classification. GeoScout-Agent uses iterative zoom-based inspection to distinguish hydrological erosion patterns from anthropogenic disturbance, correctly attributing slope failure to precipitation-driven processes overlooked by single-pass inference.

