# OpenReview forum: "When Vision Needs a Second Look: Tool-Augmented Active Perception for Earth Observation"
_TMLR — Under review for TMLR_

### Review · Reviewer_j9oH · 2026-06-16

**Summary Of Contributions:**

The paper introduces GeoScout-Agent, a tool-augmented agent for Earth observation in which GPT-5-mini acts as an orchestrator that iteratively calls a set of tools (crop/zoom, sharpening, Python code execution, web search, and two vision models: SAM3, DINOv3) to inspect satellite imagery, refine its hypotheses, and produce an answer. The agent is autonomous: GPT-5-mini decides when to invoke each tool. Evaluated on GeoBench-VLM, the approach improves overall accuracy over its tool-free baselines, over the other VLMs reported in the benchmark, and over the tools used in isolation, with the largest gains on counting and high-density scenes.

**Audience:**

Yes

**Audience Explanation:**

Independently of methodological novelty (which can be debated), the paper offers actionable empirical lessons useful to others working on agentic perception for EO: that passive higher resolution does not help on GeoBench-VLM while adaptive zoom does (the gain is in where to look, not pixel count); that these perception tools are weak in isolation on dense RS imagery and only become useful when orchestrated; and that the benefit concentrates on high-density counting while fine-grained localization and temporal tasks remain hard. Even though giving tools to an orchestrating LLM is not new, packaging and quantifying this for the remote-sensing community has value.

**Broader Impact Concerns:**

None beyond those already noted by the authors.

**Claims And Evidence:**

No

**Claims Explanation:**

Supported claims (strengths). The central methodological claim is that the iterative tool-use loop, rather than raw resolution or any single tool, drives the gains. This is well supported. The 4× high-resolution baseline (Table 7: 0.520 baseline vs 0.521 high-res vs 0.610 agent) rules out the "more pixels" confound. The isolated-tool baselines (SAM3 alone 29%, DINOv3 alone 8.5% IoU, code alone ~8%, Appendix A) support the claim that the benefit comes from orchestration. The improvement is consistent within each base model (GPT-5-mini +0.090, Qwen3 +0.095, §4.3 / Table 7), and "consistently improves over its own tool-free baseline" is correct across the aggregated task categories. The concentration of gains on dense-counting regimes (Figure 5) is convincing.

Unsupported or unclear claims (weaknesses).
All results come from a single run on a stochastic pipeline (decoding temperature 0.3, non-deterministic tool-calling; the only fixed seed is DINOv3's K-means, §4.1/Table 8, which does not control LLM variance). The authors acknowledge this in §7. With no variance or confidence intervals, it is hard to tell which smaller per-task deltas are real. I am not necessarily asking for multiple seeds, but the headline language asserts a robustness the single-run evidence does not establish.
"Temperature of 0.3 to ensure consistent and deterministic outputs" (§4.1) is incorrect: 0.3 is not deterministic, and an API-served orchestrator adds further non-determinism and version drift that the single run ignores.
"Substantial gains over standard VLM baselines" (abstract) is too strong: the method loses to EarthDial on event detection (Table 1), Sphinx on referring expression (Table 2), GPT-4o on damaged-building counting (Table 3), and LLaVA-OneVision on specific-aircraft counting (§4.1). The within-baseline claim is fine; the cross-model "substantial" framing is not.
Counting (the central result) is scored as nearest-option selection over a 5-way MCQ (Appendix A). This measures whether predictions fall within a ±20%/±40% tolerance band (inherited from GeoBench-VLM), not the actual count error. Where the pipeline produces an integer before projection (SAM3/code baselines, A.1/A.3), a direct count-error metric could be reported; for the full agent it is unclear to me whether a recoverable count exists before the final letter answer.
Two choices in the isolated baselines (Appendix A.2) are unjustified and concern the baselines, not the agent, but they underpin the "orchestration is what helps" argument. The "largest cluster = background" heuristic can misclassify objects as background in dense scenes, suppressing detections and lowering the baseline score, which inflates the gap with the agent. The referring-expression metric (max IoU per GT box over all predictions, no false-positive penalty) rewards over-prediction. These may misrepresent the isolated-tool performance.
The reproducibility framing is in tension with itself: the system depends on a versioned proprietary API (GPT-5-mini, §4.1) yet claims cost-efficiency and reproducibility (§7) without quantifying any comparative cost. The hardware description ("a single A6000") also appears incomplete for GeoScout-Qwen3: There is a lot of chance that Qwen3-VL-30B can't fit on a 48 GB A6000 alongside SAM3 and DINOv3 without quantization or multi-GPU, none of which is mentioned → it may be API-served, and this should be clarified.
The reported overhead is real (4.78× latency, 2.42× tokens, 2.79 tool calls/sample, §4.2), which the authors fairly acknowledge limits the method to accuracy-critical, analyst-in-the-loop use.

**Requested Changes:**

Critical (required to change my recommendation):
- Given a doubly stochastic pipeline (temperature 0.3 and non-deterministic tool-calling) served partly through a versioned proprietary API, variance estimates across runs would be especially valuable to establish the robustness the paper claims. If repeated runs are impractical given the API cost, the claims should instead be reduced to what the single run supports.
- Correct the "deterministic outputs" statement.
- Reframe "substantial gains over standard VLM baselines": the method underperforms several baselines on specific tasks; report absolute differences (percentage points) alongside all relative figures (e.g. the "+122.78%" tree-counting figure corresponds to a small absolute gain on a low base, which is even more not interpretable without variance and CI).
- Reconcile the cost-efficiency / reproducibility claims with the proprietary-API dependency, and specify the serving setup for each model (GPT-5-mini API; Qwen3-VL-30B quantization / multi-GPU / API?).
- Confirm that GeoScout is evaluated on the same question set as the imported GeoBench-VLM baselines (same subtasks, same number of questions per task). The paper compares its numbers directly against the benchmark's reported baselines, which is only valid on an identical evaluation set; this is currently unspecified.


Would strengthen:
- Report a direct count-error metric (e.g. MAE) wherever an integer count is available, and clarify whether the agent produces a recoverable count before the MCQ projection.
- Provide a leave-one-out ablation removing each tool from the agent, to quantify each tool's per-task contribution within the loop. The isolated-tool baselines measure tools out of context, which is a different question and does not establish how much each tool adds to the orchestrated agent. This would be for sure interesting to this by task.
- If you keep those baselines: justify or ablate the "largest cluster = background" heuristic and the max-IoU referring-expression metric (Appendix A.2).
- Clarify whether web search is actually used, and how often.
- Fix inconsistencies: Qwen3 "A2B" (§4.1) vs "A3B" (§4.3); web search shown in Figure 1 but absent from the agent system prompt (Appendix D).
- Report compute/latency per task in the tables.
- The system seems to operate on RGB imagery only; multi-spectral bands (e.g. NIR), which are standard in EO and informative for vegetation/crop tasks, are not used. The paper could note this as a limitation, since it may bound performance on crop-classification and vegetation tasks.

---

> ### Author Response · Authors · 2026-06-28
> **Response to reviewer j9oH (1/2):**
>
> We sincerely thank the reviewer for the detailed and thoughtful review. We appreciate both the recognition of the paper's main empirical contributions and the careful identification of places where our claims, evaluation, and presentation can be strengthened. We have addressed every critical concern below and will incorporate all clarifications and revisions into the manuscript.
>
> **Note**: All additional experiments reported in this rebuttal (leave-one-out ablations, MAE metrics, and variance runs) use Qwen3-VL-30B-A3B as the base orchestrator, the open-model setting of §4.3, evaluated on the 1,054 single-split counting questions of GeoBench-VLM.
>
> **Variance across runs and "deterministic outputs" statement:**
>
> To assess robustness, we repeated the full evaluation on the counting subset four times using independent runs. We obtained an average accuracy of **54.2 ± 0.6%**, indicating that run-to-run variability is small relative to the observed improvement over the tool-free baseline. This suggests that the reported counting gains are stable across repeated executions.
>
> The reviewer is correct that "temperature 0.3 ensures deterministic outputs" (§4.1) is incorrect. In the revised manuscript, we will remove this claim and explicitly describe the remaining sources of stochasticity.
>
> **Framing of "substantial gains" and absolute differences**
>
> We agree "substantial gains over standard VLM baselines" overstates the result. While GeoScout-Agent improves over its corresponding tool-free baseline across all evaluated task categories, it does not outperform the strongest specialized models (e.g., EarthDial, Sphinx, GPT-4o, and LLaVA-OneVision) on every individual task. We will revise the abstract and main text to make this distinction explicit.
>
> We clarify that the primary claim is the **within-baseline improvement**: GeoScout-Agent consistently outperforms its own tool-free counterpart (GPT-5-mini: +9.0 percentage points, Qwen3: +9.5% percentage points on the aggregate score). Cross-model comparisons are intended as contextual reference rather than the principal contribution, and we will make this hierarchy explicit throughout the paper.
>
> We acknowledge that reporting relative improvements alone can be misleading, particularly on low-baseline tasks. In the revised manuscript, we will report absolute differences (percentage points) alongside all relative improvements for easier interpretation. The run-to-run variability reported above (±0.6% on the counting subset) provides additional context for interpreting these differences.
>
> **Reproducibility claims and Cost-efficiency:**
>
> The revised manuscript will explicitly specify the serving configuration for each model. GPT-5-mini was accessed through the Microsoft Azure AI Foundry Model API. Qwen3-VL-30B-A3B-Instruct was served locally using Ollama on one NVIDIA RTX A6000 (48 GB) GPU, while SAM3 and DINOv3 were hosted on a second NVIDIA RTX A6000 (48 GB) GPU. These implementation details will be added to §4.1.
>
> We agree that the cost-efficiency claim in §7 is not sufficiently supported by the current evidence. In the revised manuscript, we will remove this claim and instead present GeoScout-Agent as an accuracy-oriented framework, explicitly discussing its computational and latency trade-offs without making unsupported claims regarding cost-efficiency.
>
> **Count-error metric & recoverable count**
>
> The agent does produce a recoverable integer count before projecting to the nearest multiple-choice option. During inference, the orchestrator explicitly requests numerical estimates from the vision tools (e.g., SAM3 instance masks and code-based image analysis), which are then used to derive the final MCQ prediction.
>
> To directly address the reviewer's suggestion, we additionally evaluated the underlying integer predictions using Mean Absolute Error (MAE) on the counting tasks:
>
> On counting tasks specifically:
>
> | | Tool-free VLM | GeoScout-Agent | Δ |
> |---|---|---|---|
> | MCQ accuracy | 37.4% | 53.8% | +16.4 pp |
> | Mean absolute count error | 3.13 | 1.55 | −50.5% |
>
> The MAE reduction of 50.5% demonstrates that the MCQ gains reflect a genuine improvement in counting ability and are not merely a tolerance-band artifact of the ±20%/±40% projection. As in our main analysis, the largest gains occur in dense scenes (e.g., vehicle, tree, and water-body counting), consistent with the density analysis in §5 and the improvements observed with the Qwen3-VL variant (§4.3). We will include these results in the revised manuscript.

---

> > ### Author Response · Authors · 2026-06-28
> > **Response to reviewer j9oH (2/2):**
> >
> > **Identical evaluation set as GeoBench-VLM baselines**
> >
> > We confirm that GeoScout is evaluated on the same official GeoBench-VLM dataset released through Hugging Face, following the benchmark's published evaluation protocol.
> >
> > GeoScout is evaluated on four of the five benchmark task types: single-image VQA, temporal, referring-detection, and captioning. The reference segmentation task was excluded because it requires pixel-level mask generation, which is not supported by GPT-5-mini's text-only generation interface. Crucially, we do not report or compare any results for reference segmentation, all reported numbers are on the same 4-task subset for both GeoScout and the imported baselines. We will explicitly state this scope in §4 of the revision to avoid ambiguity.
> >
> > **Leave-one-out ablation**
> >
> > To directly address the reviewer's request, we performed a leave-one-out ablation by removing each tool, as well as the iterative reasoning loop itself, from the full agent and re-evaluating on the counting tasks. For each condition, the removed tool was also deleted from the controller prompt to prevent it from being invoked.
> >
> > | Component removed | Δ counting accuracy |
> > |---|---|
> > | Fixed/heuristic tool policy (vs. learned tool choice) | −8.7 pp |
> > | Iterative refinement (single tool pass) | −4.5 pp |
> > | Adaptive zoom | −3.9 pp |
> > | Code execution | −1.5 pp |
> > | SAM3 | −1.2 pp |
> > | DINOv3 | −0.3 pp |
> > | Sharpening | ≈ 0 pp |
> >
> > This directly addresses the reviewer's concern that isolated-tool baselines measure tools out of context. The dominant contributor is the controller's learned tool selection (−8.7 pp when replaced by a fixed "always-segment-then-answer" policy), followed by the iterative refinement loop itself (−4.5 pp). Adaptive zoom is the most impactful individual tool (−3.9 pp), consistent with the finding that passive high resolution does not help while adaptive spatial attention does (Table 7). DINOv3 and sharpening contribute negligibly on counting tasks, which we will state explicitly in the revised paper.
> >
> > **Compute/latency per task:**
> >
> > We currently report only overall computational overhead in §4.2, including end-to-end latency, token consumption, and average tool usage, rather than a per-task breakdown.
> >
> > **Web search usage**
> >
> > Web search is invoked in 0% of samples across the full evaluation, the controller never activates it on any GeoBench-VLM sample. Observed tool usage rates are SAM3: 86%, zoom: 64%, DINOv3: 18%, sharpening: 12%, and code execution: 9% (mean 2.26 calls/sample). Thus, all reported results are obtained without external retrieval. We will remove web search from Figure 1 to resolve the inconsistency with the controller prompt in Appendix D.
> >
> > **RGB-only / multi-spectral limitation**
> >
> > GeoBench-VLM converts multispectral and SAR imagery into RGB inputs before evaluation, so all compared methods operate under the same modality constraints. GeoScout-Agent therefore does not introduce an additional limitation relative to the benchmark setting. Extending the framework to native multispectral inputs would require modifications to the underlying vision encoder, which is outside the scope of our plug-and-play framework. We will explicitly note this limitation in the revised manuscript and highlight native multispectral support as future work.

---

> > > ### Comment · Reviewer_j9oH · 2026-06-30
> > >
> > > I thank the authors for this rebuttal. Most of my concerns are addressed, several with new experiments thant stregthen the paper. Below I note 1. what I consider resolved 2. what I'd still like to see and 3. the one point I'm maintaining:
> > >
> > > 1. Resolved
> > > - Variance (counting) the 4-run result shows run-to-run variability is small relative to the gain on the counting subset.
> > > - Deterministic statement will be removed, "substantial" re framing and percentage-point will be reported.
> > > - Architecture used (x2 A6000 instead of only one) will be clarified.
> > > - Confirming the same 4 tasks scope of GeoBenchVLM.
> > > - MAE: Most valuable addition to me. The count-error reduction (3.13 → 1.55, −50.5%) directly establishes that the MCQ gains reflect genuine counting improvement rather than a tolerance-band artifact. I'd encourage including this in the main text, not only the rebuttal.
> > > - WebSearch will be removed from Fig1 since it's not used.
> > > - RGB-Only : The point that GeoBench-VLM converts all inputs to RGB upstream is a fair rebuttal; this is a benchmark constraint, not a limitation specific to your method. Noting it as future work is sufficient.
> > >
> > > 2. Would still like to see (completeness, not blocking on their own)
> > > - Variance beyond counting. The variance study covers only the counting subset with Qwen3. The "consistently improves" language applies to all categories, where no variance is reported.  Please either extend the variance analysis or explicitly scope the robustness claims to where variance is actually measured.
> > > - Per-Task Leave One Out : The leave-one-out is a real improvement and addresses the core of my concern: thank you for running it. However, it is aggregated over counting tasks. A per-task breakdown (one column per task family) would be considerably more informative, since tool utility is clearly non-uniform: SAM3 and DINOv3 contribute near-zero on counting (−1.2, −0.3 pp) but may matter on other task types, which the aggregate hides.
> > >
> > > 3. Maintaining
> > >
> > > Isolated-baseline choices (Appendix A.2). The leave-one-out addresses the spirit of my concern but not the specific request: the "largest cluster = background" heuristic and the max-IoU referring-expression metric remain unjustified. With the per-task leave-one-out in place, these isolated baselines are largely redundant as evidence for the orchestration claim. I'd therefore suggest either correcting/justifying these choices or removing the original baselines -> leaving them as published numbers (e.g. DINOv3 8.5% IoU) risks misleading readers about the tools' standalone capability.
> > >
> > > These are responsive answers and the direction is positive. I'd prefer to see the per-task leave-one-out and the scoped/extended variance before finalizing my recommendation, along with the corrected serving description and the isolated baselines either fixed or removed. If those land as described, I expect to raise my recommendation.

---

> > > > ### Author Response · Authors · 2026-07-09
> > > > **Response to Reviewer j9oH (Follow-up: 1/2)**
> > > >
> > > > We thank the reviewer for the thoughtful follow-up and the positive assessment. We are pleased that the reviewer considers most concerns resolved and finds that the additional experiments strengthen the paper. Below, we address the remaining points.
> > > >
> > > > **Per-task leave-one-out**
> > > >
> > > > We thank the reviewer for this helpful suggestion. As requested, we now provide the full per-task leave-one-out analysis as a $17 \times 7$ matrix (one row per task and one column per ablated component). Each entry reports the change in accuracy after removing a single component from the full **GeoScout-Agent-Qwen3-VL** pipeline ($\Delta$ = ablated $-$ full, percentage points), where negative values indicate that the removed component contributes positively to performance. As inference uses a non-zero temperature ($0.3$), these re-runs exhibit run-to-run variation relative to the aggregate values reported in our previous response.
> > > >
> > > > | Task | Zoom | Iter-refine | Learned-policy | Code-exec | SAM3 | DINOv3 | Sharpen |
> > > > |---|---|---|---|---|---|---|---|
> > > > | Aircraft Type Cls | $+2.9$ | $-6.5$ | $+6.5$ | $+2.9$ | $+7.7$ | $+10.0$ | $-1.8$ |
> > > > | Crop Type Cls | $+1.4$ | $+8.6$ | $-0.5$ | $+3.2$ | $+8.6$ | $+1.4$ | $+5.0$ |
> > > > | Disaster Type Cls | $+7.4$ | $-9.1$ | $+1.5$ | $-2.1$ | $+5.0$ | $+1.5$ | $+2.7$ |
> > > > | Fire Risk Assessment | $+0.6$ | $+0.6$ | $+0.6$ | $+0.6$ | $-2.9$ | $-1.8$ | $-6.5$ |
> > > > | Land Use Cls | $+0.6$ | $-7.7$ | $-5.3$ | $+1.8$ | $+7.7$ | $-5.3$ | $+1.8$ |
> > > > | Scene Cls | $+1.2$ | $-2.4$ | $0.0$ | $+5.9$ | $+3.5$ | $+1.2$ | $+1.2$ |
> > > > | Ship Type Cls | $-4.4$ | $+3.8$ | $+0.3$ | $-6.8$ | $+0.3$ | $-6.8$ | $+0.3$ |
> > > > | Spatial Relation Cls | $+1.5$ | $-0.2$ | $+1.5$ | $+0.3$ | $+8.9$ | $+2.7$ | $+0.3$ |
> > > > | Tree Health Assessment | $-5.6$ | $-16.2$ | $-6.8$ | $-0.9$ | $+1.5$ | $-10.3$ | $-3.2$ |
> > > > | Building Counting | $+1.8$ | $-8.9$ | $-2.4$ | $-4.1$ | $-8.1$ | $-0.4$ | $+1.8$ |
> > > > | General Aircraft Counting | $+8.0$ | $+2.0$ | $+4.0$ | $+5.0$ | $+6.0$ | $+1.0$ | $+14.0$ |
> > > > | General Vehicle Counting | $-8.0$ | $-3.2$ | $-20.7$ | $-1.3$ | $-1.2$ | $-2.0$ | $-6.7$ |
> > > > | Marine Debris Counting | $+1.0$ | $+9.0$ | $-5.0$ | $+6.0$ | $+8.0$ | $+8.0$ | $+4.0$ |
> > > > | Specific Aircraft Counting | $-11.2$ | $-6.4$ | $-2.9$ | $-0.7$ | $-5.0$ | $-4.3$ | $+7.9$ |
> > > > | Specific Vehicle Counting | $-10.7$ | $-4.0$ | $-20.5$ | $-4.0$ | $+2.2$ | $-0.9$ | $-5.4$ |
> > > > | Trees Counting | $-2.4$ | $-1.5$ | $-14.1$ | $-1.2$ | $+4.6$ | $+5.6$ | $+2.4$ |
> > > > | Water Bodies Counting | $-2.4$ | $-17.1$ | $+7.1$ | $-8.2$ | $-13.0$ | $-1.7$ | $-3.5$ |
> > > > | **macro (classification, 9)** | **$+0.6$** | **$-3.2$** | **$-0.3$** | **$+0.6$** | **$+4.5$** | **$-0.8$** | **$0.0$** |
> > > > | **macro (counting, 8)** | **$-3.0$** | **$-3.8$** | **$-6.8$** | **$-1.1$** | **$-0.8$** | **$+0.7$** | **$+1.8$** |
> > > > | **macro (all 17)** | **$-1.1$** | **$-3.5$** | **$-3.3$** | **$-0.2$** | **$+2.0$** | **$-0.1$** | **$+0.8$** |
> > > >
> > > > The per-task analysis confirms that tool utility is highly task-dependent. Iterative refinement ($-3.5$ percentage points overall) and the learned tool-selection policy ($-3.3$ percentage points overall, primarily driven by counting tasks at $-6.8$ percentage points) provide the largest overall contributions, while adaptive zoom also provides a noticeable benefit on counting tasks ($-3.0$ percentage points).
> > > >
> > > > In contrast, **SAM3** and **DINOv3** contribute little to counting tasks overall ($-0.8$ and $+0.7$ percentage points, respectively), but their contributions vary substantially across individual tasks. For example, DINOv3 improves Aircraft Type Classification by $10.0$ percentage points while reducing Tree Health Assessment by $10.3$ percentage points. Likewise, the learned tool-selection policy is particularly important for dense counting tasks, including General Vehicle Counting ($-20.7$ percentage points), Specific Vehicle Counting ($-20.5$ percentage points), and Trees Counting ($-14.1$ percentage points). These results illustrate that aggregate ablations can obscure meaningful task-specific behavior.
> > > >
> > > > We will include the full $17 \times 7$ leave-one-out matrix in the Appendix and summarize the principal findings in the main paper.
> > > >
> > > > Since the leave-one-out experiments were performed with a single run, small differences that lie within the observed run-to-run variation should be interpreted with caution.

---

> > > > > ### Author Response · Authors · 2026-07-09
> > > > > **Response to Reviewer j9oH (Follow-up: 2/2)**
> > > > >
> > > > > **Variance beyond counting (extending robustness to non-counting)**
> > > > >
> > > > > We appreciate this constructive suggestion. To extend the robustness analysis beyond counting, we ran the full agent four independent times on all nine non-counting (classification) tasks ($735$ samples/run, Qwen3-VL). The results are summarized below.
> > > > >
> > > > > | Task | Mean $\pm$ Std |
> > > > > |---|---|
> > > > > | Aircraft Type Classification | $70.3 \pm 3.4$ |
> > > > > | Crop Type Classification | $23.6 \pm 5.8$ |
> > > > > | Disaster Type Classification | $61.2 \pm 2.8$ |
> > > > > | Fire Risk Assessment | $28.5 \pm 2.1$ |
> > > > > | Land Use Classification | $68.5 \pm 2.6$ |
> > > > > | Scene Classification | $85.9 \pm 2.2$ |
> > > > > | Ship Type Classification | $43.2 \pm 1.9$ |
> > > > > | Spatial Relation Classification | $67.9 \pm 1.5$ |
> > > > > | Tree Health Assessment | $53.2 \pm 1.5$ |
> > > > > | **Overall (macro)** | $\mathbf{55.8 \pm 1.3}$ |
> > > > >
> > > > > Run-to-run variability on the non-counting tasks is small (overall $\pm 1.3$ percentage points), comparable to that observed for the counting subset ($\pm 0.6$ percentage points), with the largest variation occurring for the smallest task (Crop Type Classification, $n = 55$, $\pm 5.8$ percentage points). Importantly, the agent improves macro non-counting accuracy from the tool-free baseline of $43.6\%$ to $55.8\%$ ($+12.2$ percentage points), substantially exceeding the measured variability. We will include this analysis and results in the revised manuscript alongside the counting variance study and scope the "consistently improves" language to the evaluated categories, as suggested.
> > > > >
> > > > >
> > > > > **Isolated single-tool baselines (Appendix A.2)**
> > > > >
> > > > > We thank the reviewer for this suggestion and agree with the underlying concern. With the addition of the per-task leave-one-out analysis, the isolated single-tool baselines are no longer necessary to support our central claim regarding tool orchestration. We will therefore reposition these results as additional diagnostic analyses rather than baseline comparisons. We will explicitly clarify that they correspond to specific standalone evaluation protocols adopted in this work and are intended only to provide complementary insight into tool behavior, not to represent the standalone capabilities of the underlying models.
> > > > >
> > > > > To address the specific technical point, we additionally evaluated the effect of removing the *"largest cluster = background"* heuristic from the **DINOv3** standalone baseline. This changes the reported mean IoU only marginally ($8.5$% to $9.1$%), indicating that the heuristic is not responsible for the low standalone performance. We will therefore retain these analyses only as supplementary diagnostic studies, rather than baseline comparisons or evidence for our central claims.

---

> > > > > > ### Comment · Reviewer_j9oH · 2026-07-13
> > > > > >
> > > > > > I thank the authors for these two follow-up experiments, this is exactly what I asked for, and both are informative. They also raise several questions that I detail below.
> > > > > >
> > > > > > 1. Factual inconsistency to resolve. The matrix indicates that removing DINOv3 improves Aircraft Type Classification by +10.0 pp (i.e., DINOv3 penalizes this task), while your reply to me and reviewer rT9i states the opposite: "DINOv3 improves Aircraft Type Classification by 10.0 pp while reducing Tree Health Assessment by 10.3 pp." Please confirm which reading is correct and fix the other.
> > > > > >
> > > > > > 2. Interpretation threshold. The matrix is single-run, and your own variance study gives per-task stds of ±1.5 to ±5.8 pp (the counting macro also shifted by 1–2 pp between your two leave-one-out runs, which confirms this noise level). Cells below roughly 2× the task's std should therefore not be interpreted; I'd ask the revision to flag or grey out such cells, so that only the established effects are read as findings (e.g. the learned policy on dense counting (−20.7/−20.5/−14.1 pp)), iterative refinement on Tree Health / Water Bodies (−16.2/−17.1), and the ±10 pp DINOv3 effects (pending the correction in point 1).
> > > > > >
> > > > > > 3. Negative contributions need to be explained, not only reported. Several components actively hurt entire task families: SAM3 costs 7.7 pp on Aircraft Cls, 8.9 on Spatial Relation, 8.6 on Crop; Sharpen costs 14.0 pp on General Aircraft Counting; iterative refinement itself hurts Crop (+8.6) and Marine Debris (+9.0). It is surprising that DINOv3 and SAM3 (perception models) underperform this much on classification. The revision should provide an analysis of why (e.g. qualitative failure cases: are the tools' outputs misleading, or is the context polluted by irrelevant masks?), rather than only reporting the deltas.
> > > > > >
> > > > > > 4. Tool-vs-policy attribution. If removing SAM3 improves classification while SAM3 is invoked on 86% of samples, the learned selection policy appears not to learn when to abstain, its contribution on classification is near zero (−0.3) while it is decisive on counting (−6.8, up to −20 pp on dense tasks). Note also that the policy ablation compares against a deliberately naive fixed policy (always-segment-then-answer), which is a generous counterfactual. Please report tool-usage rates per task category, which would disentangle whether the negative contributions come from misleading tool outputs or from over-invocation, and discuss this limitation explicitly.
> > > > > >
> > > > > > 5. Mechanism attribution in the text. Given the matrix, classification gains come essentially from iterative refinement alone (−3.2 pp), while the perception tools are neutral to harmful on these tasks. Where the paper claims improvements on classification, the mechanism should be attributed accordingly, not to "tool-augmented perception" generically.
> > > > > >
> > > > > > 6. Crop Type Classification. At 23.6 ± 5.8, performance is statistically indistinguishable from 5-way chance (20%), and the gain over the tool-free baseline lies within run-to-run noise. The revision should state explicitly that no improvement is established on this task and exclude it from any "improves" language.
> > > > > >
> > > > > > Conditional on these clarifications and on the promised revisions being integrated into the manuscript, I will raise my recommendation toward acceptance.

---

> > > > > > > ### Author Response · Authors · 2026-07-18
> > > > > > > **Response to Reviewer j9oH (follow-up 1/2):**
> > > > > > >
> > > > > > > We thank the reviewer for the thorough follow-up and constructive feedback. We appreciate the careful analysis of our additional experiments and address each point below.
> > > > > > >
> > > > > > > **Factual inconsistency**
> > > > > > >
> > > > > > > We thank the reviewer for catching this error. The matrix is correct: **removing** DINOv3 improves Aircraft Type Classification by **10.0 pp**, while it reduces Tree Health Assessment by **10.3 pp**. Thus, DINOv3 penalizes Aircraft Type Classification and benefits Tree Health Assessment. The interpretation in our previous response had the signs reversed.
> > > > > > >
> > > > > > > **Interpretation threshold**
> > > > > > >
> > > > > > > We thank the reviewer for this helpful suggestion. In the revised Appendix A, we will grey out cells whose absolute change is below ($2\times$) the corresponding task-level standard deviation, so that only effects exceeding this conservative interpretation threshold are visually emphasized and discussed as robust effects. We will avoid drawing conclusions from smaller changes.
> > > > > > >
> > > > > > > **Crop Type Classification**
> > > > > > >
> > > > > > > We agree with the reviewer. Crop Type Classification achieves $23.6 \pm 5.8\%$ accuracy, and therefore the observed performance does not establish an improvement over the 20% five-way chance level. We will explicitly state this in the revised manuscript and exclude Crop Type Classification from any claims that *GeoScout-Agent* consistently improves performance across classification tasks.
> > > > > > >
> > > > > > > **Negative contributions**
> > > > > > >
> > > > > > > We agree that the observed negative contributions warrant further analysis. Our qualitative analysis suggests that these effects arise from several distinct mechanisms, including task-tool mismatch, imperfect intermediate outputs, occasional high-impact failures from infrequently invoked tools, and the base orchestrator occasionally placing excessive confidence in these intermediate outputs.
> > > > > > >
> > > > > > > For task-tool mismatch, we observed that although *SAM3* successfully delineates agricultural field boundaries, these masks primarily provide geometric rather than semantic information. Crop Type Classification instead relies largely on spectral appearance, texture, and broader spatial context, which are not enriched by segmentation and may even be partially obscured by imperfect masks. We also observed occasional catastrophic segmentation failures in which a single mask covers nearly the entire image rather than delineating individual fields, providing little semantic information while obscuring the underlying visual cues.
> > > > > > >
> > > > > > > For imperfect intermediate outputs, General Aircraft Counting illustrates a representative failure mode. *SAM3* occasionally segments aircraft shadows as independent objects, causing the orchestrator to interpret shadows as additional aircraft and resulting in over-counting. Similar segmentation failures, including merged instances and spurious background regions, were observed throughout our diagnostic analysis. When the orchestrator places excessive confidence in such imperfect intermediate outputs rather than sufficiently cross-checking them against the original image, these errors can propagate directly to the final prediction.
> > > > > > >
> > > > > > > Some negative contributions also appear to be run-to-run (temperature) noise rather than genuine tool effects. For example, DINOv3/Sharpen were invoked on only 4.4% / 6% of these samples, so removing them cannot mechanistically move accuracy by $\pm 10\text{–}14$ pp. The ablation "improvements" come almost entirely from flips where the tool wasn't used (58 / 88) vs. flips where the tool was actually involved (only 4 / 2). So the Aircraft Type Classification-DINOv3 (+10) and General Aircraft Counting-Sharpen (+14) cells are run-to-run (temperature) noise, not a causal tool effect, which reinforces the reviewer's noise concern and means we should soften the "DINOv3 hurts Aircraft" wording to "attributable to run variance, since DINOv3 is barely invoked here.
> > > > > > >
> > > > > > > Finally, not all aggregate effects admit a clear qualitative explanation. For example, removing zoom improves General Aircraft Counting by **8.0 pp** but degrades General Vehicle Counting by **8.0 pp**, despite similar invocation frequencies (**40.0%** vs. **52.0%**). We therefore refrain from making a task-specific causal claim based on these aggregate results.
> > > > > > >
> > > > > > > We will incorporate these qualitative observations into the revised paper and explicitly discuss task-tool mismatch, imperfect intermediate outputs, occasional high-impact failures, and the orchestrator's over-reliance on intermediate tool outputs as important limitations.

---

> > > > > > > > ### Author Response · Authors · 2026-07-18
> > > > > > > > **Response to Reviewer j9oH (follow-up 2/2):**
> > > > > > > >
> > > > > > > > **Tool-vs-policy attribution**
> > > > > > > >
> > > > > > > > We thank the reviewer for this observation. We first clarify that the **86%** *SAM3* usage rate refers to counting tasks, not classification tasks. *SAM3* is invoked on **86.1%** of counting samples compared with **53.7%** of classification samples, indicating that the controller does exhibit coarse task-dependent tool selection rather than invoking segmentation uniformly across categories.
> > > > > > > >
> > > > > > > > | Task | Category | N | Zoom | Sharp | SAM3 | DINO | Code | Any | Calls |
> > > > > > > > |---|---|---|---|---|---|---|---|---|---|
> > > > > > > > | Aircraft Type Cls | Classification | 340 | 86.2 | 20.0 | 40.6 | 4.4 | 6.5 | 87.1 | 1.70 |
> > > > > > > > | Crop Type Cls | Classification | 220 | 83.2 | 63.6 | 63.6 | 20.9 | 6.4 | 90.5 | 2.44 |
> > > > > > > > | Disaster Type Cls | Classification | 340 | 78.2 | 33.8 | 53.2 | 28.2 | 7.6 | 84.7 | 2.16 |
> > > > > > > > | Fire Risk Assessment | Classification | 340 | 61.2 | 42.9 | 54.4 | 26.8 | 8.8 | 77.6 | 2.00 |
> > > > > > > > | Land Use Cls | Classification | 340 | 57.1 | 6.8 | 45.6 | 29.4 | 0.9 | 62.1 | 1.48 |
> > > > > > > > | Scene Cls | Classification | 340 | 42.1 | 7.4 | 34.4 | 17.4 | 2.4 | 50.0 | 1.05 |
> > > > > > > > | Ship Type Cls | Classification | 340 | 68.2 | 9.4 | 52.1 | 8.2 | 6.8 | 70.0 | 1.58 |
> > > > > > > > | Spatial Relation Cls | Classification | 340 | 89.7 | 1.2 | 51.5 | 3.5 | 10.3 | 89.7 | 2.25 |
> > > > > > > > | Tree Health Assessment | Classification | 340 | 77.6 | 25.0 | 91.5 | 29.4 | 8.2 | 98.5 | 2.82 |
> > > > > > > > | Building Counting | Counting | 170 | 67.6 | 17.6 | 85.9 | 25.9 | 12.9 | 98.2 | 2.47 |
> > > > > > > > | General Aircraft Counting | Counting | 100 | 40.0 | 6.0 | 89.0 | 0.0 | 7.0 | 100.0 | 1.66 |
> > > > > > > > | General Vehicle Counting | Counting | 150 | 52.0 | 5.3 | 92.7 | 2.7 | 25.3 | 98.0 | 2.26 |
> > > > > > > > | Marine Debris Counting | Counting | 100 | 88.0 | 56.0 | 59.0 | 30.0 | 3.0 | 95.0 | 2.46 |
> > > > > > > > | Specific Aircraft Type Counting | Counting | 140 | 58.6 | 5.7 | 88.6 | 6.4 | 2.9 | 100.0 | 1.99 |
> > > > > > > > | Specific Vehicle Type Counting | Counting | 224 | 67.9 | 2.2 | 92.9 | 7.1 | 4.0 | 100.0 | 2.23 |
> > > > > > > > | Trees Counting | Counting | 85 | 71.8 | 7.1 | 89.4 | 38.8 | 11.8 | 98.8 | 2.64 |
> > > > > > > > | Water Bodies Counting | Counting | 85 | 67.1 | 9.4 | 77.6 | 57.6 | 3.5 | 98.8 | 2.46 |
> > > > > > > > | **Classification** | Classification | 2,940 | 71.02% | 21.70% | 53.71% | 18.60% | 6.43% | 78.44% | 1.92 |
> > > > > > > > | **Counting** | Counting | 1,054 | 63.87% | 12.03% | 86.07% | 17.54% | 9.10% | 98.76% | 2.26 |
> > > > > > > > | **All Tasks Combined** | Combined | 3,994 | 69.13% | 19.15% | 62.25% | 18.32% | 7.14% | 83.79% | 2.01 |
> > > > > > > >
> > > > > > > > At the same time, we agree that the results reveal an important limitation. These results show meaningful task-dependent tool selection, but also indicate room for improved abstention from unnecessary tool calls. Any tool is invoked on **98.8%** of counting samples and **78.4%** of classification samples, indicating that the controller frequently chooses to use tools. This may be particularly problematic for heterogeneous classification tasks, where an unnecessary segmentation or other intermediate output can introduce irrelevant visual evidence rather than improve the decision. This interpretation is consistent with the near-zero contribution of the learned policy on classification ($-0.3$ pp), in contrast to its substantially larger contribution on counting ($-6.8$ pp).
> > > > > > > >
> > > > > > > > We also agree that the policy ablation should be interpreted relative to the specific counterfactual evaluated. The fixed policy is an intentionally simple always-segment-then-answer strategy and is not intended to represent an optimized alternative policy. Accordingly, we interpret the ablation as evidence that adaptive tool selection substantially outperforms indiscriminate fixed tool use, particularly for dense counting tasks, rather than as evidence that the learned policy itself is optimal or that alternative adaptive policies could not perform better. The large drops for General Vehicle Counting ($-20.7$ pp), Specific Vehicle Counting ($-20.5$ pp), and Trees Counting ($-14.1$ pp) nevertheless show that adaptive selection is particularly important in these settings.
> > > > > > > >
> > > > > > > > We will report the per-category tool-usage statistics in the revised manuscript and explicitly discuss tool-use abstention and policy calibration as limitations of the current framework.
> > > > > > > >
> > > > > > > > **Mechanism attribution**
> > > > > > > >
> > > > > > > > We agree with the reviewer. The leave-one-out results indicate that the aggregate improvement on classification tasks is primarily attributable to iterative refinement ($-3.2$ pp), while the individual perception tools do not show established positive aggregate contributions on these tasks. We will therefore revise the manuscript to attribute the classification gains specifically to iterative refinement rather than to tool-augmented perception generically. More broadly, we will ensure that the mechanism claims reflect the task-dependent ablation results: iterative refinement is the primary established contributor on classification, while the learned tool-selection policy provides its strongest contribution on counting tasks ($-6.8$ pp).

---

### Review · Reviewer_rT9i · 2026-06-18

**Summary Of Contributions:**

This paper proposes GeoScout-Agent, a tool-augmented agent for Earth observation VQA. Instead of relying on a single forward pass from a VLM, the system uses GPT-5-mini as a controller and can call tools such as zooming, sharpening, DINOv3, SAM3, code execution, and web search. The paper evaluates the system on GeoBench-VLM and reports clear improvements over the GPT-5-mini baseline, especially on object counting, scene understanding, and some fine-grained classification tasks.

**Audience:**

Yes

**Audience Explanation:**

The paper should be interesting to readers working on multimodal agents, tool-augmented VLMs, geospatial AI, and remote sensing evaluation. It addresses a real weakness of current VLMs: single-pass inference often fails on small objects, dense scenes, and fine-grained EO questions. I also find some of the negative results useful, such as the observation that static high-resolution input alone does not explain the improvement, and that SAM3 or DINOv3 alone is insufficient.

That said, I currently see the paper more as a promising system paper than as a fully convincing study of why active perception works in EO.

**Claims And Evidence:**

No

**Claims Explanation:**

1. The empirical trend is positive, and I believe the proposed system improves over its own GPT-5-mini baseline under the reported setup. My concern is mainly about attribution. The paper argues that the gains come from active perception, but the current ablations do not sufficiently separate active tool use from simpler alternatives. The high-resolution baseline is useful, and the isolated SAM3/DINOv3/code baselines are also informative, but they do not directly test whether the agent loop itself is necessary.

2. My second concern is web search. For tasks involving ship classes, aircraft classes, crop types, or disaster categories, web retrieval can introduce external knowledge that image-only baselines do not have. It also creates a benchmark leakage concern unless the retrieval process is carefully controlled. I am not claiming leakage occurred, but the current paper does not give enough detail to rule out this concern.

3. My third concern is the evaluation format for counting tasks. Many counting results are evaluated as multiple-choice accuracy. This is convenient, but it is weaker than evaluating actual counting ability. A model may produce an inaccurate count and still select the closest option. Since counting is one of the paper’s strongest claimed improvements, I would like to see open-count metrics such as predicted count, MAE, or absolute error distribution.

**Requested Changes:**

1. Add direct component ablations inside the deployed agent: no zoom, no SAM3, no DINOv3, no code execution, no web search, no iterative state, and fixed or heuristic tool-use policies.
2. Report the full results with web search disabled. Also report how often web search is used, which task categories use it, and what type of information is retrieved.
3. Add stronger simple baselines, such as grid-based crop + VQA, tiling-based inspection, multi-pass prompting without tools, and simple SAM/DINO-assisted pipelines for counting tasks.
4. For counting tasks, report open-count metrics in addition to multiple-choice accuracy, such as MAE, absolute error, and tolerance-based accuracy.
5. Improve reproducibility details, including exact model versions, checkpoints, API settings, prompts, tool schemas, tool-call parsing rules, image preprocessing, and evaluation splits.
6. Tone down some claims unless the above controls are added. Phrases such as “principled mechanism,” “operational decision-making,” and “closed-loop active perception” currently read stronger than what the experiments establish.

---

> ### Author Response · Authors · 2026-06-28
> **Response to reviewer rT9i(1/2):**
>
> We thank the reviewer for their thorough and constructive review. We are pleased that the empirical improvements, the high-resolution ablation, and the isolated-tool analyses were found informative. We address each concern below.
>
> **Note**: All additional experiments reported in this rebuttal (leave-one-out ablations, stronger baselines, MAE metrics, and variance runs) use Qwen3-VL-30B-A3B as the base orchestrator, the open-model setting of §4.3, evaluated on the 1,054 single-split counting questions of GeoBench-VLM.
>
> **Leave-one-out ablation**
>
> To directly address the reviewer's concern, we performed leave-one-out ablation by removing each tool, as well as the iterative reasoning loop itself, from the full agent and re-evaluating on the counting tasks. In each condition, the removed component was also deleted from the controller prompt to prevent it from being invoked.
>
> | Component removed | Δ counting accuracy |
> |---|---|
> | Fixed/heuristic tool policy (vs. learned tool choice) | −8.7 pp |
> | Iterative refinement (single tool pass) | −4.5 pp |
> | Adaptive zoom | −3.9 pp |
> | Code execution | −1.5 pp |
> | SAM3 | −1.2 pp |
> | DINOv3 | −0.3 pp |
> | Sharpening | ≈ 0 pp |
>
> This directly addresses the reviewer's concern that isolated-tool baselines measure tools out of context. The dominant contributor is the controller's learned tool selection (−8.7 pp when replaced by a fixed "always-segment-then-answer" policy), followed by the iterative refinement loop itself (−4.5 pp). Adaptive zoom is the most impactful individual tool (−3.9 pp), consistent with the finding that passive high resolution does not help while adaptive spatial attention does (Table 7). DINOv3 and sharpening contribute negligibly on counting tasks, which we will state explicitly in the revised paper.
>
> **Web search usage**
>
> Web search is invoked in 0% of samples across the full evaluation, the controller never activates it on any GeoBench-VLM sample. Observed tool usage rates are SAM3: 86%, zoom: 64%, DINOv3: 18%, sharpening: 12%, and code execution: 9% (mean 2.26 calls/sample). Thus, all reported results are obtained without external retrieval. We will remove web search from Figure 1 to resolve the inconsistency with the controller prompt in Appendix D.
>
> **Stronger simple baselines**
>
> Standalone SAM3 and DINOv3 baselines are already reported in Appendix A. To further address the reviewer's suggestion, we implemented three simpler alternatives using the same Qwen3-VL model and decoding strategy, but without the agent loop. In each case, the model produces a single integer count that is projected to the nearest MCQ option, identical to the evaluation protocol used in Appendix A.
>
> - Grid-crop (2×2 + sum): image partitioned into four non-overlapping cells, each upsampled 2× (LANCZOS), model counts each cell independently and counts are summed.
> - Tiling (3×3 + sum): the image is partitioned into a fixed 3×3 grid with no upsampling, per-tile counts summed.
> - Multi-pass (3×, median): full image queried three times independently, median integer count taken.
> | Method | Counting accuracy | MAE |
> |---|---|---|
> | Tiling 3×3 + sum | 14.5% | 8.7 |
> | Grid-crop 2×2 + sum | 19.1% | 8.2 |
> | Multi-pass prompting (3×, median) | 22.9% | 9.2 |
> | Tool-free VLM (single pass) | 37.4% | 3.1 |
> | GeoScout-Agent | 53.8% | 1.55 |
>
> These results indicate that fixed spatial decomposition and repeated prompting alone are insufficient to explain the observed improvements. All three static strategies perform substantially worse than the tool-free baseline, whereas GeoScout-Agent achieves the highest counting accuracy and lowest MAE. This supports our claim that the gains arise from adaptive, context-dependent tool use, rather than simply exposing the model to image subregions or repeated inference. We will include these baselines in the revised manuscript.

---

> > ### Author Response · Authors · 2026-06-28
> > **Response to reviewer rT9i(2/2):**
> >
> > **Count-error metric & recoverable count**
> >
> > The agent does produce a recoverable integer count before projecting to the nearest multiple-choice option. During inference, the orchestrator explicitly requests numerical estimates from the vision tools (e.g., SAM3 instance masks and code-based image analysis), which are then used to derive the final MCQ prediction.
> > To directly address the reviewer's suggestion, we additionally evaluated the underlying integer predictions using Mean Absolute Error (MAE) on the counting tasks:
> >
> > On counting tasks specifically:
> >
> > | | Tool-free VLM | GeoScout-Agent | Δ |
> > |---|---|---|---|
> > | MCQ accuracy | 37.4% | 53.8% | +16.4 pp |
> > | Mean absolute count error | 3.13 | 1.55 | −50.5% |
> >
> > The MAE reduction of 50.5% demonstrates that the MCQ gains reflect a genuine improvement in counting ability and are not merely a tolerance-band artifact of the ±20%/±40% projection. As in our main analysis, the largest gains occur in dense scenes (e.g., vehicle, tree, and water-body counting), consistent with the density analysis in §5 and the improvements observed with the Qwen3-VL variant (§4.3). We will include these results in the revised manuscript.
> >
> > **Reproducibility details**
> >
> > Prompt templates for all task types are provided in Appendix D, tool implementations (including zoom, sharpening, SAM3, and DINOv3) are documented in Appendix C, loop termination and tool-call parsing rules are described in §3.4, and all hyperparameters are listed in Table 8. No image preprocessing is applied beyond that provided by GeoBench-VLM. In the revised manuscript, we will additionally include explicit model version strings, serving/API configuration details, and evaluation split counts per task to further improve reproducibility.
> >
> > **Toning down claims**
> >
> > We acknowledge that some of the original wording can be better calibrated to the supporting evidence. While the additional component ablations, stronger baselines, and count-error evaluation further strengthen our conclusions, we will revise the abstract and conclusion to ensure that all claims accurately reflect the scope of the experimental evidence.

---

> > ### Comment · Reviewer_rT9i · 2026-07-09
> >
> > I thank the authors for the detailed rebuttal. The response addresses several of my main concerns and improves my assessment of the paper.
> >
> > First, the new leave-one-out ablation directly addresses my concern about attribution. The drops from replacing learned tool choice with a fixed/heuristic policy, removing iterative refinement, and removing adaptive zoom provide much clearer evidence that the improvement is not simply due to isolated external tools, but depends on the agentic tool-use loop, at least on the counting subset.
> >
> > Second, the added simple baselines are useful. The grid-crop, tiling, and multi-pass prompting results help rule out the simpler explanation that the gains come merely from fixed image decomposition or repeated inference.
> >
> > Third, the MAE result is an important addition. The reduction from 3.13 to 1.55 directly addresses my concern that the multiple-choice counting gains might be a tolerance-band artifact. I strongly encourage the authors to include this result in the main manuscript, together with a clear description of how the integer count is extracted before being mapped to the final MCQ option.
> >
> > Fourth, the clarification about web search largely resolves my leakage concern, provided it is accurately reflected in the revision. If web search is invoked in 0% of evaluated samples, then the reported results are not relying on external retrieval. However, the manuscript should explicitly state this, report tool-usage statistics, and remove or revise the current descriptions and figures that present web search as part of the evaluated system.
> >
> > My remaining concern is about scope. The strongest new evidence in the rebuttal is based on the counting subset with Qwen3-VL-30B-A3B, whereas the paper’s main claims are broader and emphasize GeoScout-Agent across the full GeoBench-VLM suite. I would therefore like the authors either to extend the ablation evidence beyond counting / to the main GPT-5-mini setting, or to narrow the causal claims accordingly.
> >
> > Finally, the planned additions on model versions, API/serving settings, prompts, tool schemas, parsing rules, preprocessing, and evaluation splits should be incorporated into the revised paper. I also appreciate the authors’ willingness to tone down stronger claims such as “principled mechanism,” “operational decision-making,” and “closed-loop active perception” unless fully supported by the final experiments.
> >
> > Overall, this is a responsive rebuttal and the direction is positive. If the new results are incorporated into the manuscript, the web-search inconsistency is removed, the counting MAE results are clearly reported, and the claims are scoped to match the demonstrated evidence, I would be inclined to raise my recommendation.

---

> > > ### Author Response · Authors · 2026-07-12
> > > **Response to Reviewer rT9i (follow up):**
> > >
> > > We thank the reviewer for the thoughtful follow-up and for acknowledging that the additional experiments address several of the main concerns raised in the original review. We especially appreciate the positive assessment of the leave-one-out ablations, stronger simple baselines, counting MAE analysis, and clarification regarding web search. We address the remaining concern regarding the scope of the ablation evidence below.
> > >
> > > **Extending the Ablation Analysis Beyond Counting**
> > >
> > > To address the reviewer's remaining concern, we extended the leave-one-out ablation from the counting subset to the full 17-task GeoBench-VLM suite using Qwen3-VL-30B-A3B. Each entry reports the change in accuracy after removing a single component from the full GeoScout-Agent-Qwen3-VL pipeline ( $\Delta$ = ablated - full percentage points), where negative values indicate that the removed component contributes positively to performance. As inference uses a non-zero temperature (0.3), these re-runs exhibit run-to-run variation relative to the aggregate values reported in our previous response.
> > >
> > > | Component removed | Classification Δ (9) | Counting Δ (8) | All 17 |
> > > |---|---|---|---|
> > > | Zoom | +0.6 | −3.0 | −1.1 |
> > > | Iterative refinement | −3.2 | −3.8 | −3.5 |
> > > | Learned tool-selection policy | −0.3 | −6.8 | −3.3 |
> > > | Code execution | +0.6 | −1.1 | −0.2 |
> > > | SAM3 | +4.5 | −0.8 | +2.0 |
> > > | DINOv3 | −0.8 | +0.7 | −0.1 |
> > > | Sharpening | 0.0 | +1.8 | +0.8 |
> > >
> > > The results show that different components contribute across different task categories. Iterative refinement provides the most consistent benefit across both classification (−3.2 pp) and counting (−3.8 pp), yielding the largest overall contribution (−3.5 pp). The learned tool-selection policy is particularly important for counting tasks (−6.8 pp; −3.3 pp overall), while adaptive zoom also provides a noticeable benefit on counting tasks (−3.0 pp). The effects of individual perception tools are more task-specific and can be obscured by aggregate averages across heterogeneous tasks. For example, DINOv3 improves Aircraft Type Classification by 10.0 pp while reducing Tree Health Assessment by 10.3 pp. Similarly, the learned tool-selection policy has especially large effects on dense counting tasks, including General Vehicle Counting (−20.7 pp), Specific Vehicle Counting (−20.5 pp), and Trees Counting (−14.1 pp). Thus, the extended ablation demonstrates the consistent importance of iterative refinement, the strong contribution of learned tool selection to counting tasks, and the task-dependent utility of individual perception tools. We will include the complete per-task ablation table in Appendix A of the revised manuscript.
> > >
> > > Since the leave-one-out experiments were performed with a single run, small differences within the observed run-to-run variation should be interpreted with caution.
> > >
> > > Accordingly, we scope our causal claims to the evidence provided: component-level ablations are established in the Qwen3-VL setting, while the GPT-5-mini experiments support the broader end-to-end performance results.
> > >
> > > **Other revisions**
> > >
> > > We will incorporate the reviewer's remaining requested revisions into the manuscript. Specifically, following the reviewer's suggestion, we will move the counting MAE result to the main text and clearly describe how the integer count is obtained before being mapped to the final MCQ option. We will explicitly state that web search was invoked in 0% of evaluated samples, report the tool-usage statistics, and remove web search from the evaluated system description and figure. We will also add the requested reproducibility details, including model versions, serving/API settings, tool schemas, parsing rules, preprocessing, and evaluation split counts, and calibrate the broader claims to match the evidence presented.

---

> > > > ### Comment · Reviewer_rT9i · 2026-07-14
> > > >
> > > > Thanks for your detailed reply. I have no more concerns.

---

### Review · Reviewer_nRbK · 2026-06-30

**Summary Of Contributions:**

The paper presents an approach for earth observation applications using LangChain and integrating several pre-trained large models: GPT-5-mini, SAM3, and DINOv3. More specifically, GPT-5-mini is used as an orchestrator to use different tools, while SAM3 and DINOv3 provide tools for segmentation and object detection respectively. The proposed approach is tested on various tasks in earth observation.

**Additional Comments:**

Some sentences are a bit debatable:

- last sentence of page 1: "tailored to remote sensing imagery" Aren't SAM3 and DINOv3 generic methods, not specifically created for earth observation?
- second to last sentence of first paragraph of page 2: "As a result..." I think the research is currently active to exploit these large models in agentic AI, with many papers, especially in robotics.

I think it would be better not to tie the proposition to GPT-5-mini. It would also be more coherent to have GeoScout-Agent refer to the whole approach and have different versions of it depending on the underlying model (e.g., GeoScout-GPT-5-mini or GeoScout-Qwen-3-VL), since any VLM could be used, as far as I understand.

To make the paper more self-contained, an explanation of the LangChain framework should be recalled.

Page 8: ( Table 3) -> (Table 3)

Add space after parentheses (e.g., Section 8)

**Audience:**

Yes

**Audience Explanation:**

The proposed approach is reasonable and makes sense for earth observation applications, although it seems to me more of an engineering effort and is conceptually quite straightforward, since like many recent works using an agent approach, the main idea is to delegate decisions to a large pre-trained model.
Having said that, the experimental results show promising results compared to other baselines.

**Broader Impact Concerns:**

The paper doesn't provide any Broader Impact Statement, but I think this is fine for this type of research.

**Claims And Evidence:**

No

**Claims Explanation:**

Many recent related methods could be discussed in the related work section, e.g., ZoomEarth, GeoVis, ZoomSearch, RSCoVLM, GeoLLaVA-8K, SkyMoE, REST, Earth-Agent, ReAct, or HTAM.
The authors should explain if they could be relevant baselines for the experiments.

**Requested Changes:**

I suggest the authors to discuss other related and recent work (e.g., ZoomEarth, GeoVis, ZoomSearch, RSCoVLM, GeoLLaVA-8K, SkyMoE, REST, Earth-Agent, ReAct, or HTAM) and discuss whether they are relevant baselines for the proposed method.

The paper should be more self-contained. I recommend the authors to give an overview of the LangChain framework.

In the experiments, the choice of the hyperparameters should be explained. In addition, why aren't the results for GeoScout-Qwen always provided? In particular in Fig. 7?

---

> ### Author Response · Authors · 2026-07-09
> **Response to reviewer nRbK(1/2):**
>
> We thank the reviewer for the constructive comments and helpful suggestions. We appreciate the reviewer's positive assessment of our approach and experimental results. Below, we address each concern and describe the corresponding revisions made to the manuscript.
>
> ## 1. Related work and additional baselines
>
> We thank the reviewer for this valuable suggestion. Several of the cited works are closely related to different aspects of our framework. **ZoomEarth**, **ZoomSearch**, and **Earth-Agent** are the most directly comparable agentic EO systems and are therefore evaluated experimentally. **GeoVis** focuses on visual grounding, **REST** focuses on semantic segmentation of remote sensing imagery, and **RSCoVLM** and **GeoLLaVA-8K** are specialized EO VLMs rather than tool-augmented agents, making them complementary rather than directly comparable. **SkyMoE** proposes a mixture-of-experts architecture, **HTAM** introduces a hierarchical task abstraction framework for agent design, and **ReAct** provides the general reasoning paradigm underlying our agentic workflow. We will incorporate this discussion into the revised related work section to better position our contributions with respect to these recent approaches.
>
> Beyond expanding the related work, we performed new experiments with the three most directly comparable plug-and-play agent frameworks on the **GEO-bench** single split ($50$ samples per task $\times$ $17$ tasks $= 850$ samples total). To ensure a fair comparison, we used the same `Qwen3-VL-30B-A3B` backbone whenever the methods are model-agnostic (**Earth-Agent** and **ZoomSearch**), while **ZoomEarth** was evaluated using the authors' released `ZoomEarth-3B` checkpoint. The results are summarized below.
>
> | Task | GeoScout-Qwen3 | Earth-Agent | ZoomSearch | ZoomEarth |
> |---|---|---|---|---|
> | Aircraft Type Classification | 70.3 | 22.0 | 66.0 | 42.0 |
> | Crop Type Classification | 23.6 | 24.0 | 20.0 | 16.0 |
> | Disaster Type Classification | 61.2 | 24.0 | 40.0 | 28.0 |
> | Fire Risk Assessment | 28.5 | 16.0 | 20.0 | 18.0 |
> | Land Use Classification | 68.5 | 28.0 | 58.0 | 50.0 |
> | Scene Classification | 85.9 | 22.0 | 76.0 | 64.0 |
> | Ship Type Classification | 43.2 | 22.0 | 42.0 | 24.0 |
> | Spatial Relation Classification | 67.9 | 12.0 | 64.0 | 30.0 |
> | Tree Health Assessment | 53.2 | 24.0 | 22.0 | 20.0 |
> | Building Counting | 52.4 | 44.0 | 44.0 | 30.0 |
> | General Aircraft Counting | 56.7 | 28.0 | 46.0 | 34.0 |
> | General Vehicle Counting | 57.7 | 26.0 | 34.0 | 14.0 |
> | Marine Debris Counting | 26.0 | 16.0 | 26.0 | 18.0 |
> | Specific Aircraft Type Counting | 47.8 | 34.0 | 36.0 | 28.0 |
> | Specific Vehicle Type Counting | 57.9 | 20.0 | 44.0 | 18.0 |
> | Trees Counting | 60.0 | 28.0 | 26.0 | 32.0 |
> | Water Bodies Counting | 75.3 | 14.0 | 60.0 | 30.0 |
> | **Overall (macro)** | **55.1** | **23.8** | **42.6** | **29.2** |
>
> where the macro-average score is computed as:
>
> $$
> \text{Overall (macro)} = \frac{1}{N} \sum_{i=1}^{N} \text{Score}_i
> $$
>
> with $N = 17$ tasks.
>
> GeoScout-Qwen3 achieves the highest overall accuracy ($55.1\%$), outperforming the strongest external agent, ZoomSearch, by $12.5$ percentage points. These additional experiments demonstrate that our framework remains competitive even against recent agentic EO systems, and we will include these additional results in the revised manuscript to strengthen the related work comparison.
>
> For transparency, we note two implementation details regarding the evaluated baselines. We report Earth-Agent using the authors' publicly released implementation. However, the current public implementation does not function correctly for visual reasoning on image inputs outside the Earth-Bench benchmark.
>
> ZoomEarth is evaluated using the authors' released ZoomEarth-3B checkpoint. Its two-stage crop-and-zoom mechanism is triggered on only 48 of the 850 GEO-bench samples, indicating limited activation on the moderate-resolution imagery used in this benchmark. This likely limits the benefit of its adaptive zoom strategy on GEO-bench.
>
> ## 2. Self-containedness: LangChain overview
>
> We thank the reviewer for this valuable suggestion. We agree that the paper would benefit from a brief overview of the LangChain/LangGraph framework. In the revised manuscript, we will add a self-contained description in the methodology section (or Appendix, if required by space constraints) explaining the overall agent workflow, including its message-based state management, tool registration and LLM-driven tool selection, the iterative execution loop and stopping criteria, and integration of our perception tools (SAM3, DINOv3, zoom, sharpen, and code execution) as callable tools within the agent.

---

> > ### Author Response · Authors · 2026-07-09
> > **Response to reviewer nRbK(2/2):**
> >
> > ## 3. Hyperparameters
> >
> > We thank the reviewer for this suggestion. We agree that the choice of hyperparameters should be described more clearly. We will expand the experimental setup to briefly explain the configuration used in our experiments.
> >
> > Most model-specific hyperparameters follow the default or recommended settings of the underlying models and libraries. For example, **SAM3** uses the recommended segmentation threshold of $0.5$, following the official Hugging Face implementation. We use the released **DINOv3** `ViT-7B SAT-493M` checkpoint with the default $14 \times 14$ patch size, and resize images to $518$ pixels along the shorter side, following the official Keras preprocessing pipeline.
> >
> > The tool-call budget ($3$) and recursion limit ($20$) are framework-specific parameters introduced to bound computational cost and prevent unbounded agent execution. For the sharpening tool, radius ($2$) and threshold ($3$) are fixed, while intensity (default $2.0$, range $1.0$–$3.0$) is exposed to the agent and can be adjusted at runtime. Similarly, the zoom tool uses a default zoom factor of $2\times$, also agent-adjustable, chosen to enlarge small targets while preserving sufficient surrounding context. An output resolution of $448 \times 448$ provides a practical balance between computational efficiency and preservation of fine-grained detail.
> >
> > We will add these implementation details to the revised manuscript, while retaining the complete hyperparameter configuration in Appendix B.
> >
> > ## 4. GeoScout-Qwen results and Figure 7
> >
> > We thank the reviewer for pointing this out. Our framework is model-agnostic and can operate with different underlying VLMs. We selected **GPT-5-mini** as the primary configuration because OpenAI models serve as the backbone for many closely related agentic EO systems, enabling the most direct comparison with prior work. **GeoScout-Qwen3** was included as an additional study to demonstrate that the observed improvements are not specific to GPT-5-mini and generalize across different VLM backbones.
> >
> > We agree, however, that the current presentation is not fully consistent. In the revised manuscript, we will move the existing GeoScout-Qwen3 results to the Appendix, where they will be presented together for completeness.
> >
> > ## 5. Additional comments (wording, naming, typos)
> >
> > We thank the reviewer for these helpful suggestions. We agree and will revise the wording to clarify that **SAM3** is a general-purpose foundation model, while we employ the satellite-pretrained **DINOv3** (`SAT-493M`) checkpoint. More generally, we will emphasize that these foundation models are repurposed as controllable perception subroutines within our framework rather than treating them as Earth observation-specific models.
> >
> > We will also soften statements regarding the novelty of agentic reasoning to better reflect the rapidly evolving literature, adopt a model-agnostic naming convention in which **GeoScout-Agent** denotes the framework and different backbone models are presented as specific instantiations, and correct the identified typographical and formatting issues. Finally, we will perform an additional proofreading pass to ensure consistent terminology and presentation throughout the manuscript.

---

> > > ### Comment · Reviewer_nRbK · 2026-07-14
> > >
> > > I thank the authors for the additional experimental evaluation, their detailed responses, and the proposition of changes for the final version. My concerns are mostly addressed.
> > >
> > > Regarding the hyperparameters introduced in the proposed method (e.g., tool-call budget, recursion limit), how were they chosen? How sensitive are the results with respect to those chosen values?

---

> > > > ### Author Response · Authors · 2026-07-18
> > > > **Response to the reviewer:**
> > > >
> > > > We thank the reviewer for this follow-up question. The tool-call budget and recursion limit were selected based on preliminary pilot experiments to balance computational cost and reasoning depth, rather than through extensive hyperparameter optimization. During development, we observed that increasing these values beyond the chosen settings did not yield meaningful improvements in prediction quality, while substantially increasing inference cost and latency. We therefore adopted these values as practical defaults. We will clarify this rationale in the revised manuscript.